# Distribution of cross-tropopause convection within the Asian monsoon region from May through October 2017

Corey E. Clapp[1], Jessica B. Smith[1], Kristopher M. Bedka[2], James G. Anderson[1,3,4]

[1]Harvard John A. Paulson School of Engineering and Applied Sciences, Harvard University, Cambridge, 02138, U.S.A.
[2]NASA Langley Research Center, Hampton, 23681, U.S.A.
[3]Department of Chemistry and Chemical Biology, Harvard University, Cambridge, 02138, U.S.A.
[4]Department of Earth and Planetary Sciences, Harvard University, Cambridge, 02138, U.S.A.

*Correspondence to*: Corey E. Clapp (cclapp@fas.harvard.edu)

**Abstract.** We constructed a database of cross-tropopause convection in the Asian monsoon region for the months of May through October of 2017 using overshooting tops (OTs), deep convective features that penetrate the local cirrus anvil layer and the local tropopause, with Meteosat-8 geostationary satellite detections. The database of 40,918 OTs, represents a hemispheric record of convection covering the study domain from 10°S to 55°N and from 40 to 115°E. With this database, we analyzed the geographic, monthly, and altitude distribution of this convection and compared it to the convective distributions represented by satellite observations of outgoing longwave radiation (OLR) and precipitation. We find that cross-tropopause convection is most active during the months of May through August (with daily averages of these months above 300 OTs/day) and declines through September and October. Most of this convection occurs within North and South India, the Bay of Bengal, and the Indian Ocean regions, which together account for 75.1% of all OTs. We further identify distinct, differing seasonal trends within the study subregions. For the North India, South India, and Bay of Bengal regions the distribution of OTs follows the development of the Asian Monsoon, with its north-south movement across the study period. This work demonstrates that when evaluating the effects of convection on lower stratospheric composition over the Asian monsoon region it is important to consider the impact of cross-tropopause convection specifically, as well as the contributions from both land-based and oceanic regions due to the significant geographic and monthly variation in convective activity.

## 1 Introduction

Deep convection in the Asian monsoon region provides an important pathway for the transport of boundary layer air to the stratosphere (e. g. Brunamonti et al., 2018; Khaykin et al., 2022; Lelieveld et al., 2018; Park et al., 2007; Randel et al., 2010; Santee et al., 2017). Which areas within the Asian monsoon region, however, contribute the most to this convective transport remain uncertain. Moreover, while the current understanding of convective transport considers deep convection that reaches the upper troposphere, less attention has been paid to the role of cross-tropopause convection (Legras & Bucci,

2020; Tissier & Legras, 2016). To investigate the potential impacts of cross-tropopause convection on convective transport and the composition of the lower stratosphere we construct a seasonal database of "overshooting tops," (OTs) deep convective updrafts that penetrate both the cirrus anvil layer and the local tropopause, in the Asian monsoon region from geostationary satellite infrared imagery for the period of May through October of 2017. With this database we identify the primary source regions of cross-tropopause convection as well as the seasonal and altitude trends of this deep convection. Understanding the convective transport within the Asian monsoon is critical to understanding the transport of both radiatively and chemically active species from the boundary layer to the lower stratosphere (Brioude et al., 2010; Claxton et al., 2019; Pisso et al., 2010; Tegtmeier et al., 2020a). In particular, the timing and location of convection determine which species in what quantities are transported upwards from the source region boundary layer to the lower stratosphere.

Deep convection associated with the Asian monsoon has been shown to influence the composition of the lower stratosphere. Convective transport of boundary layer air to the upper troposphere/lower stratosphere (UTLS) over the Asian monsoon impacts the concentrations of many chemical species that are both radiatively and chemically significant. Observations of this convective influence range from large-scale satellite studies to small scale in situ measurements and cover a wide range of chemical perturbations of tropospheric origin. To begin with, satellite observations have shown water vapor and carbon monoxide maxima as well as ozone minima concurrent with the Asian monsoon anticyclone that develops in the UTLS region (Luo et al., 2018; Park et al., 2007; Randel et al., 2010; Santee et al., 2017). Modelling studies, including chemistry-climate models (e.g. Pan et al., 2016; Wu et al., 2020) and Lagrangian-trajectory models (e.g. Legras & Bucci, 2020; Tissier & Legras, 2016), have demonstrated the convective source of these water vapor, carbon monoxide, and ozone perturbations. Further, recent in situ measurements have also verified their convective origins (Bucci et al., 2020; Johansson et al., 2020; von Hobe et al., 2021). In addition, deep convection has been shown to transport anthropogenic pollutants such as peroxyacetyl nitrate and acetylene (Johansson et al., 2020), and other non-methane hydrocarbons (Baker et al., 2011). Both satellite and in situ measurements have also indicated that deep convection is the primary source of the aerosols in the UTLS in the Asian monsoon region (e.g. Bian et al., 2020; Brunamonti et al, 2018; Hanumathu et al., 2020; Höpfner et al., 2019; Vernier et al., 2015; Vernier et al., 2018). Furthermore, this convection may transport very-short lived species (VSLS), including ozone depleting substances (ODS), to the UTLS (Adcock et al., 2021; Fiehn et al., 2018; Hossaini et al., 2016; Tegtmeier et al., 2020a).

Trajectory studies have identified radiatively driven uplift across the tropopause within the Asian monsoon anticyclone (AMA) as the primary pathway for convective influenced air to enter the lower stratosphere (e.g. Legras & Bucci, 2020; Vogel et al., 2019; Yan et al., 2019). Rapid convective transport deposits boundary layer air in the upper troposphere above the level of zero radiative heating (LZRH) and subsequent heating results in slow ascent within the AMA (Bergman et al., 2013; Chen et al., 2012; Tissier & Legras, 2016). The rate of the slow ascent in these trajectory studies depends on the reanalysis used to evaluate the wind fields (Tegtmeier et al., 2020b); for example, an ascent of approximately 1.0-1.5 K per day has been shown using ERA-Interim (e.g. Legras & Bucci, 2020). The AMA, a large anticyclonic circulation in the UTLS with a variable spatial extent that spans from northeast Africa to east Asia (e.g. Pan et al., 2016;

Randel & Park, 2006; Vogel et al., 2015), acts as a transport barrier to confine air detrained by convection (Ploeger et al., 2015; Poshyvailo et al., 2018). This results in a "spiralling ascent" over months within the AMA across the tropopause (Legras & Bucci, 2020; Vogel et al., 2019).

    The impact of tropospheric air convectively transported into the lower stratosphere, however, is not confined to the AMA. Both eastward and westward eddy shedding by the AMA has the potential to transport pollutants and boundary layer
air from the Asian monsoon region into the midlatitudes and global stratosphere (Fadnavis et al., 2018; Fujiwara et al., 2021; Garny & Randel, 2013; Popovic & Plumb, 2001; Vogel et al., 2014). In situ measurements of stratospheric air with perturbed chemical compositions (Lelieveld et al., 2018; Müller et al., 2016; Rolf et al., 2018) and lidar observations of enhanced stratospheric aerosols (Khaykin et al., 2017) sourced back to the AMA have been observed in the midlatitudes of the Northern Hemisphere.

The location and timing of deep convection influence the kind and amount of tropospheric source gases and aerosol transported into the AMA. Due to the heterogeneity of surface emissions within the Asian monsoon region, the chemical composition of convectively transported boundary layer air will vary significantly depending on the source region. Furthermore, the location of the initial convective transport determines the pathway of entry into the lower stratosphere and therefore which regions are impacted by subsequent transport within the large-scale circulation. For example, several studies
investigating the ozone depleting potentials (ODPs) of various chemical species have found that the influence of source regions and seasons alone results in significantly different ODPs for the same emissions (Brioude et al., 2010; Claxton et al., 2019; Pisso et al., 2010; Tegtmeier et al., 2020a). Therefore, understanding the distribution of deep convection within the Asian monsoon region is necessary to predict its net effect on the composition of the UTLS, both locally and globally.

    Prior work on convective influence within the Asian monsoon region has focused primarily on convective transport
from the boundary layer to upper troposphere followed by diabatic heating and ascent into the lower stratosphere. Yet, the relative importance of various source regions remains unclear. For example, modelling studies (Bergman et al., 2013; Heath & Fuelberg, 2014; Pan et al., 2016; Wu et al., 2020) and satellite observations (Fu et al., 2006) have found that deep convection over the Tibetan Plateau is a key contributor. In contrast, additional studies have identified the Bay of Bengal and the west Pacific as critical source regions using model simulations (Chen et al., 2012; James et al., 2008) and satellite
observations (Devasthale & Fueglistaler, 2010). Other studies that combine trajectory analysis with satellite observations of deep convective cloud tops have suggested that while the Tibetan Plateau may present a uniquely efficient pathway into the UTLS, it is a minority contributor compared to other source regions including the Asian mainland (Legras & Bucci, 2020; Tissier & Legras., 2016). Moreover, these convective source regions are found to have distinct tropopause-crossing regions (Chen et al., 2012). Much of this uncertainty likely arises from the difficulty in accounting for, or resolving, the small
temporal and spatial-scales necessary to represent convection. In particular, in modelling studies differences in reanalyses used and parameterizations of convection will further complicate the representation of convective transport.

    Previous studies investigating the global distribution and frequency of extreme convection using non-Sun-synchronous satellite observations have shown the Asian monsoon region to be of importance (Liu & Liu, 2016; Liu et al.,

2020; Zipser et al., 2006). This work also found that extreme convection was more likely to occur over land than over ocean,

and that the mid-latitudes had frequencies comparable to the tropics. Importantly, when considering extreme convection that reached or surpassed the tropopause, the Asian monsoon region remained an area of significant activity (Liu & Liu, 2016; Liu et al., 2020).

The research produced by the StratoClim project, which examined deep convection within the Asian monsoon region during the same time period as the present study (2017), provides important context. Aircraft in situ measurements of

water vapor (Khaykin et al., 2022; Lee et al., 2019), CO (Bucci et al., 2020; Lee et al., 2021; von Hobe et al., 2021), ozone (Johansson et al., 2020; von Hobe et al., 2021), $N_2O$ (von Hobe et al., 2021), as well as HNO3, peroxyacetyl nitrate, ethylene, and formic acid (Johansson et al., 2020) show clear evidence of convective influence on the UTLS comprising up to 100% of the air (Bucci et al., 2020). While some studies indicate that transport of convectively influenced air across the tropopause primarily occurs via slow ascent (von Hobe et al., 2021), others demonstrate that cross-tropopause convection

directly impacts the LS (Khaykin et al., 2022; Lee et al., 2019; Lee et al., 2021). As von Hobe et al. (2021) discuss, these results are not necessarily contradictory, but indicate the complexities of the microphysics during convective transport. Modelling studies of transport pathways into the UTLS show that convection that surpasses the LZRH will ultimately influence the stratosphere due to slow ascent across the tropopause (Legras & Bucci, 2020; Nützel et al., 2019; Vogel et al., 2019; Yan et al., 2019). Convection that directly crosses the tropopause, however, has a greater probability of influencing the

LS of the northern hemisphere than ascending through the tropical pipe compared to convectively influenced air that enters the stratosphere via slow ascent (Yan et al., 2019).

The present analysis builds on prior studies of the convective transport of boundary layer air into the lower stratosphere and of the frequency of intense convection over the Asian monsoon region by focusing specifically on cross-tropopause convection across a broad geographic region for a time period that covers the entire Asian monsoon. Cross-

tropopause convection is focused on as a special class of extreme convection that directly reaches the lower stratosphere bypassing slow-ascent across the tropopause and directly connecting the convective source region and the location of entry into the lower stratosphere. We accomplish this by exploiting convective features (OTs), which represent both the finest-scale and deepest convection, to assess convective transport directly into the lower stratosphere within this specific region. Using OTs identified with geostationary satellite infrared imagery (Bedka & Khoplenkov, 2016), we construct a database of

cross-tropopause convection, and describe the geographic, monthly and altitude distribution of this convection over the Asian monsoon region from May through October of 2017, a period selected to overlap with the StratoClim project. We also compare this distribution of cross-tropopause convection to the distribution of overall convection as represented by satellite observations of outgoing longwave radiation (OLR) and precipitation for the same period and region. This work is complementary to the previously discussed studies which largely focused on tropospheric convection that reaches altitudes

above the LZRH before slowly ascending into the stratosphere.

## 2 Data and Methods

### 2.1 OT database

The OT database, derived from Meteosat-8 data, represents a hemispheric record of convection covering the study domain from 10°S to 55°N and from 40 to 115°E (see Figure 1) for the time period of 01 May, 2017 through 31 October, 2017. The OTs are algorithmically identified using multispectral imagery from Meteosat-8 with a horizontal resolution of approximately 4 km (Bedka & Khlopenkov, 2016; Yost et al., 2018). Updrafts that overshoot their anvil altitude but do not reach the tropopause are not included in this study. Only OTs with detection probability ratings (as determined by the temperature differences between the OT and the anvil, the tropopause as identified with the WMO lapse-rate definition and the MERRA-2 reanalysis data, and the local level of neutral buoyancy) greater than or equal to 0.9 were used in this analysis. This threshold detected ~50% of human-identified OTs randomly sampled throughout the world (Bedka & Khlopenkov, 2016). At this rating, the false detection rate was determined to be ~10%, and these errant detections were typically found in close proximity to actual OT regions, so inclusion of these samples does not adversely affect the results. The Bedka and Khlopenkov (2016) OT identification method is a conservative methodology that underpredicts OTs allowing for high confidence in the OTs that are detected. The potential temperature of each OT was derived from the OT IR temperature and pressure derived from the OT height and MERRA-2 (Gelaro et al., 2017) reanalysis, using the method of Griffin et al. (2016).

The OT database does not represent a complete budget of cross-tropopause convection occurring in the study region and over the time period considered. This is because the Meteosat-8 OT data were acquired every 15 minutes, while the average lifetime of an OT can be as short as several minutes (Bedka & Khlopenkov, 2016). Consequently, the OT dataset used in this study represents a small percentage of the total number of OTs that occurred and should not be used to estimate total convective outflow. OTs detected in this study consist of multiple pixels identified from the Meteosat-8 multispectral imagery. The OT detection algorithm first identifies local brightness temperature minima embedded within likely anvil clouds. Then, after filtering the minima to identify possible OTs, the algorithm characterizes the mean temperature of the anvil immediately surrounding each possible OT. Pixels surrounding the OT minima that are colder than the anvil are assumed to comprise OT regions. Here, a pixel refers to the minimum spatial unit resulting from the Meteosat-8 horizontal resolution (approximately 4 km).

We report OTs as distinct events rather than by pixel count because the temporal resolution (15 minutes) is insufficient to accurately track the temporal evolution of the spatial extent of an OT. The OT database, however, is still valid for analyzing intraseasonal variability, evolution, and distribution because the consistent and frequent sampling, the long time period under study, and the large region of interest retain the major features of OT frequency, depth, and geographic distribution. This OT detection methodology has been used in prior studies of the geographic and seasonal distributions of cross-tropopause convection (Clapp et al., 2019; Clapp et al., 2021).

## 2.2 Study regions

To identify regional differences in geographic, monthly, and vertical distribution of OTs, the study domain was subdivided into 12 regions. These regions (see Figure 1) were selected to capture geographically distinct areas of cross-tropopause convection and to allow for comparison to prior work (e.g. Bergman et al., 2013; Fu et al., 2006; Heath & Fuelberg, 2014) that examined sources of convective influence on the lower stratosphere in the Asian monsoon region. Further, many of these regions have been shown to have differences in convective characteristics such as convective initiation mechanisms and seasonal dependences (Bhat & Kumar, 2015; Romatschke & Houze, 2011; Saikranthi et al., 2018; Virts & Houze, 2016). The region of a given OT was determined by the average latitude and longitude of the pixels constituting that OT.

The Tibetan Plateau region was included because prior studies have identified it as a significant source region of convective influence on the UTLS. It has a unique convective environment due to its topography and is centrally located within the Asian monsoon anticyclone (e.g. Bergman et al., 2013; Heath & Fuelberg, 2014). The southern edge of the region, between 70° E and 95° E, which separates it from the North India region, is defined by the 3 km topographic height, taken from the MERRA-2 reanalysis. Similar boundaries have been used previously (e.g. Fu et al., 2006; Heath & Fuelberg, 2014).

The North and South India regions were included to capture the primary areas of land-based Asian monsoon convection. The separate regions are necessary to distinguish the differences in seasonality, such as monsoon onset (Kajikawa et al., 2012; Walker & Bordoni, 2016), and convective character, such as the importance of orography to convective initiation in North India (Romatschke & Houze, 2011). This distinction has also been used in prior work (e.g. Fu et al., 2006; Heath & Fuelberg, 2014). North India was also identified as a potential source region of convectively transported $NH_3$ observed during StratoClim (Höpfner et al., 2019). The Bay of Bengal, Arabian Sea, and Indian Ocean regions were defined to account for oceanic monsoon convection. The Indian Ocean region captures and separates the influence of the intertropical convergence zone (ITCZ), seen in the OT distribution (Figure 2a), from the other two oceanic regions (Bay of Bengal and Arabian Sea). The Bay of Bengal region has also been previously studied as a region of significant convective influence (e.g. Devasthale et al., 2010).

The Southeast Asia, East China, and West Pacific regions were included to distinguish the eastern portion of the Asian monsoon from the subcontinental convection, which is expected to have different intraseasonal variability and large-scale drivers (Wang et al., 2001; Wei et al., 2015; Yihui & Chan, 2005).The Arabian Peninsula region was included to investigate the climatology of the high density of OTs observed over the southwest Arabian Peninsula and Ethiopia (Figure 2a). The remaining areas within the study region, in which few OTs are observed, are covered by the Northern Latitudes, and Africa regions.

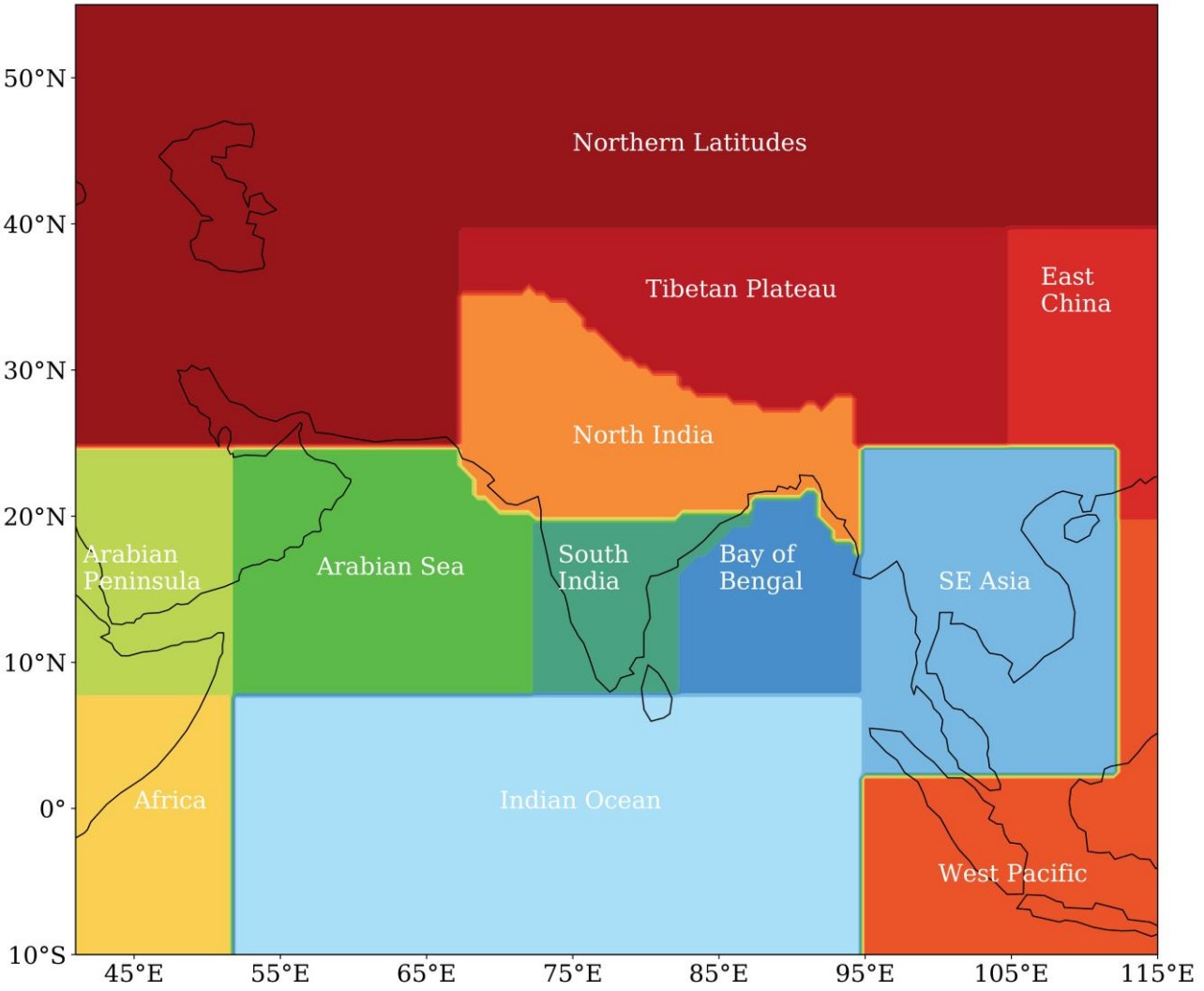

195 **Figure 1 shows the boundaries of the study area and the subdivisions of the areas used for the regional analysis.**

**2.3 Outgoing longwave radiation and precipitation data**

The OLR data is taken from the NOAA Interpolated Outgoing Longwave Radiation climatology, which has a global coverage and a resolution of 2.5° x 2.5° (Liebmann & Smith, 1996). The precipitation data is from the Global Precipitation Climatology Project (GPCP) Climate Data Record (CDR), which has global coverage and a resolution of 1° x 1° (Huffman et al., 2001).

## 3 Results

We begin with an analysis of the geographic distribution of all OTs observed across the entire study region and time period from 1 May through 31 October 2017. The distribution of cross-tropopause convection is compared with other convective indicators: OLR as a proxy for deep convection, and daily precipitation as an indicator of general tropospheric convection. The results of this analysis are then used to identify specific regions of interest for cross-tropopause convection within the larger context of monsoon convection.

### 3.1 Geographic and seasonal distribution of cross-tropopause convection

Figure 2a shows the geographic locations of all OTs across the study region for the entire study period, May through October of 2017. OTs are binned into a latitude-longitude grid of 1°x1° resolution. Areas of significant cross-tropopause convection include northern India (in particular the northwestern coast, and north coast of the Bay of Bengal), and the Bay of Bengal. The Indian Ocean shows a high volume of dispersed cross-tropopause convection located in an east-west band around the equator. In contrast, the southern edge of the Arabian Peninsula has a smaller number of OTs that are highly concentrated enough to show number densities comparable to the largest source regions (174 in the Arabian Peninsula region compared to 154 and 183 in the North India and Bay of Bengal regions, respectively). Intense convective events over the Arabian Peninsula have been previously observed (Liu & Liu, 2016; Liu et al., 2020; Zipser et al., 2006). The spatial distribution of OTs over Asia (Figure 2a) shows the importance of both land-based and oceanic convection with high frequency of cross-tropopause convection occurring in both environments.

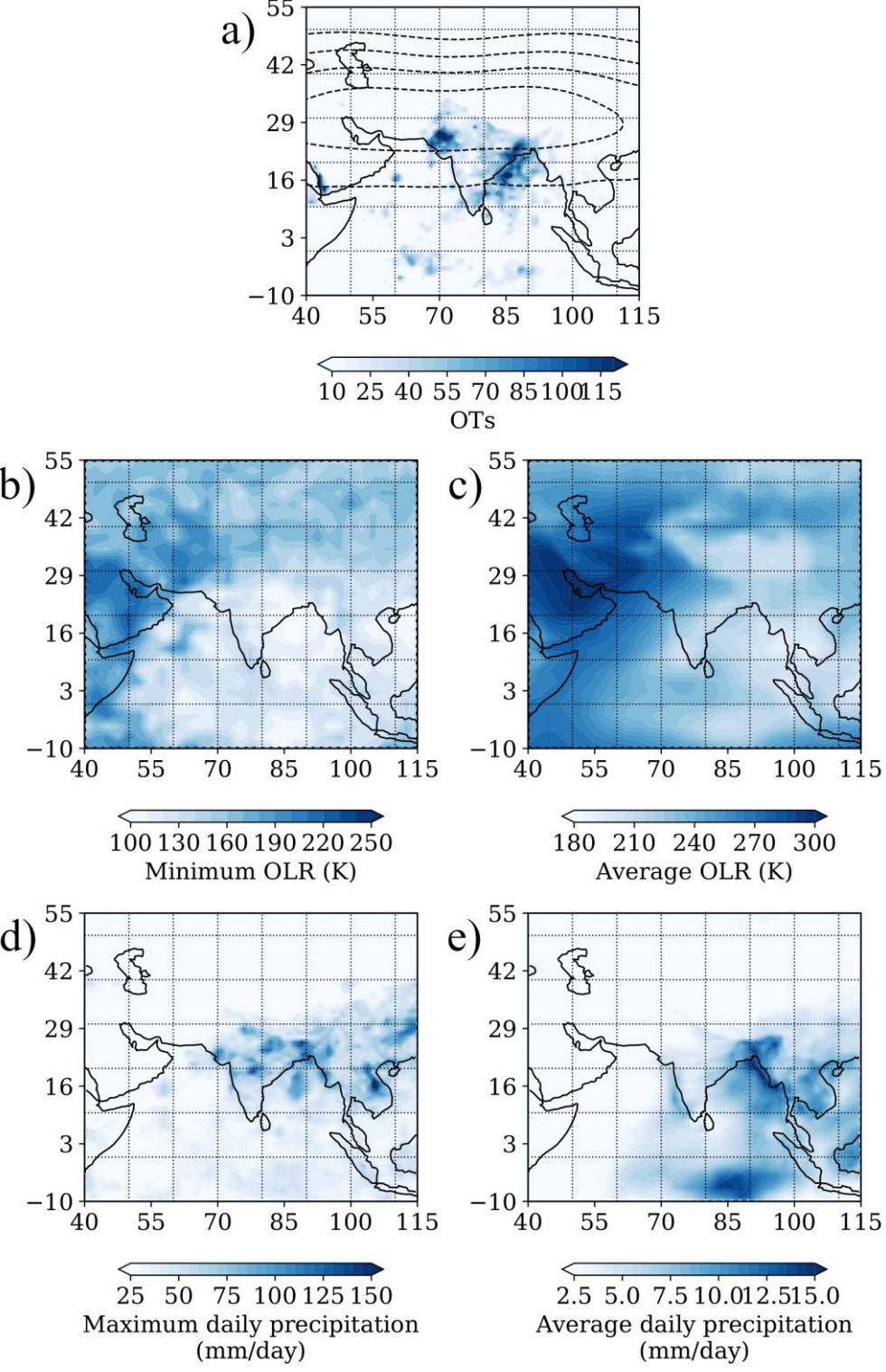

**Figure 2 The geographic distribution of convection over the Asian monsoon region during the study period (May 1, 2017 through October 31, 2017). Cross-tropopause convection is shown in Figure 2a. The colorbar shows OT frequency, evaluated in 1° x 1° bins, within the satellite observation window. Also shown is the average Montgomery potential stream-function at 400 K for the months of JJA (dashed black contours from $3.59$x$10^5$ m²s⁻² to $3.63$x$10^5$ m²s⁻² by $0.01$x$10^5$ m²s⁻²) to illustrate the location of the Asian monsoon anticyclone. The minimum and average daily OLR values are shown in Figures 2b and 2c, respectively. The minimum and average daily precipitation are shown in Figures 2d and 2e, respectively.**

Another significant feature is the absence of cross-tropopause convection occurring over the Tibetan Plateau. In contrast, a large quantity of OTs is observed over the southern slope of the plateau. Local topography defines a distinct boundary north of which few OTs occur. This is consistent with the conclusions of Tissier and Legras (2016) and Legras and Bucci (2020) in which the Tibetan Plateau is not a numerically significant contributor to convective transport, but it is found to be particularly efficient. This efficiency results from the high proportion of convective events over the Tibetan Plateau that reach the minimum level in the UT above which the slow uplift that occurs across the entire AMA will transport convectively influenced air into the LS. The lack of OT activity over the Tibetan Plateau confirms that convection in this region influences the LS through the slow uplifting process, not through cross-tropopause convection. The agreement with Legras and Bucci (2020) is particularly relevant as their study also covered the 2017 Asian summer monsoon. Similarly, the distributions observed in the global studies of extreme convection also exhibit a ridge of increased frequency along the southern slope but few convective events over the Tibetan Plateau itself (Liu & Liu, 2016; Liu et al., 2020; Zipser et al., 2006). Further, Devasthale and Fueglistaler (2010) find that while there is a significant cloud fraction over the Tibetan Plateau during July and August at 200 and 150 hPa, at 100 hPa there is almost none. Our climatology of cross-tropopause convection confirms that convection rarely transports boundary layer air directly into the lower stratosphere over the Tibetan Plateau.

Figures 2b and 2c show the OLR minimum and average daily value for the entire study period. The minimum OLR spatial distribution, as a proxy for the "deepest" convection, matches the cross-tropopause convective distribution better than the average OLR. In particular, north India, the Bay of Bengal and the Indian Ocean are regions of significant convective activity as shown by OLR. The cross-tropopause convection activity over the Arabian Peninsula visible in the OT distribution, however, is not shown to the same degree in the OLR data. The greatest disagreement between the OT distribution and the average daily OLR is present over the Tibetan Plateau, which has OLR values comparable to the other major convective regions, but very few OTs. This is likely a consequence of the unique structure of the Tibetan Plateau. The high altitude of the region results in particularly low OLR values for tropospheric convection. This could also be a consequence of frequent convection within the Tibetan Plateau region that, however, does not cross the local tropopause as often as convection in the regions marked by high OT counts.

Figures 2d and 2e show the GPCP maximum daily precipitation and average daily precipitation for the entire study period for each grid box (1° x 1°). When comparing the geographic distribution of cross-tropopause convection to the distribution of maximum daily precipitation, there is agreement over the Bay of Bengal and northern India, which are dominant regions of convection. For example, the northwest coast of the Bay of Bengal and over northwest India have "hotspots" of both high maximum daily precipitation and high densities of OTs. Further, the southern slope of the Tibetan

Plateau also forms a ridge of high precipitation values, northward of which there is less activity, similar to the sharp transition in OT frequency. The Indian Ocean also shows a band of precipitation at the equator, similar to the band of cross-tropopause convection, although of lesser magnitude. The large maximum precipitation values in central India, eastern China, and southeast Asia, however, do not correspond to OT activity, indicating that convection in this region does not reach the LS. The high concentration of convective activity visible in the OT distribution over the Arabian Peninsula is also not present in the precipitation data. This is a potential consequence of the convective events over the Arabian Peninsula being small, and therefore not detected by the relatively coarse GPCP and OLR observations.

Figure 3 displays the seasonal development of cross-tropopause convection. As seen in Figure 3a, during the active period of May through August, the daily number of OTs fluctuates largely between 200-400 OTs per day. As the monsoon season declines through September and October so does the frequency of cross-tropopause convection. For example, during the active months of May through August, on average 22.9% of the total OTs occur per month. In contrast, during the months of September and October 13.4% and 6.3% of total OTs occur, respectively. From Figure 3b, however, it is clear that different regions dominate cross-tropopause convection at different times. For example, the North India region is the most important source region during June and July, while the Bay of Bengal and Indian Ocean regions are more significant in May. As the OT database is not a complete budget of cross-tropopause convection, the values presented here quantify the contributions of different regions relative to each other, not their total convective output.

Despite different source regions having time periods of particular significance, the North India region has the highest daily number of OTs observed (see the late June and early July peaks in Figure 3b). This indicates that although June and July have similar total amounts of cross-tropopause convection to May, these months are uniquely dominated by a single source region. This is emphasized in Figure 3d, which shows that during June and July, the largest daily fractional contribution values are from North India (0.59 on June 25th and 0.64 on July 14th). This daily fractional contribution is only surpassed in October by the Bay of Bengal region. There are much fewer total cross-tropopause events during this period, however, and this peak is likely caused by specific meteorological conditions, a tropical depression within the Bay of Bengal.

From Figure 3c, the total impact of each region is visualized across these varying time periods of relative importance. North India has the greatest total amount of cross-tropopause convection (11,844 OTs, 29.0%). The Indian Ocean and Bay of Bengal regions are the next most significant contributors with 7861 OTs (19.2%) and 6212 OTs (15.2%) respectively, followed by the South India region with 4781 OTs (11.7%). Together the four largest contributors account for 75.1% of all OTs in the study area. Of the remaining OTs within the study area most occur within the Southeast Asia region (2556 OTs, 6.3%), the Arabian Sea region (2321 OTs, 5.7%), and the Arabian Peninsula region (2021 OTs, 4.9%). The Northern Latitudes region has 1919 OTs (4.7%), however, given the large size of the region this is geographically dispersed. The Tibetan Plateau and the West Pacific regions have little cross-tropopause convection with 628 OTs (1.5%) and 582 OTs (1.4%) respectively. The remaining regions of East China and Africa have 99 OTs and 94 OTs respectively, less than 1.0% combined. The different contributions to cross-tropopause convection of each region are summarized in Table 1.

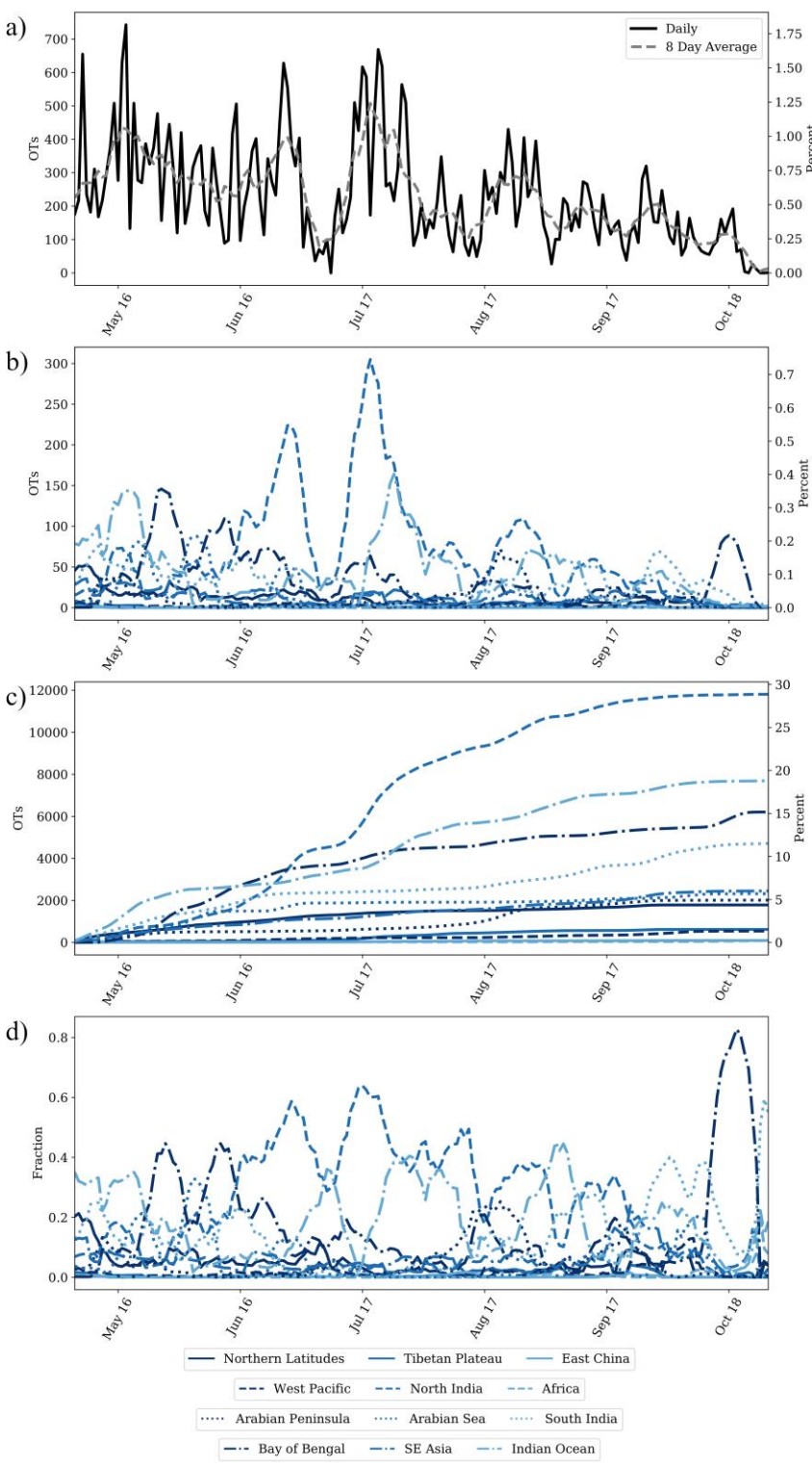

 **Figure 3 The seasonal development of cross-tropopause convection within the study region. Figure 3a shows the time series of the total daily number (left axis) and percent (right axis) of OTs in black as well as the rolling 8 day average in grey (dashed). Figure 3b shows the time series of the daily absolute number (left axis) and percent (right axis) of OTs observed within each region, as indicated by color and line style. Figure 3c shows the cumulative total number (left axis) and percent (right axis) of OTs within each region, as indicated by color and line style, as a function of time. Figure 3d shows the daily-normalized fractional contribution of each region, as indicated by color and line style, as a function of time.**

**Table 1** Summary of the regional distribution of cross-tropopause convection including the total number of OTs observed throughout the entire study period, the percentage of total OTs, and the area normalized percent contribution for each region.

| Region | OTs | Fraction [%] | Area normalized [%] |
|---|---|---|---|
| North India | 11844 | 29.0 | 29.1 |
| Indian Ocean | 7861 | 19.2 | 4.16 |
| Bay of Bengal | 6212 | 15.2 | 27.9 |
| South India | 4781 | 11.7 | 23.6 |
| Southeast Asia | 2556 | 6.25 | 4.19 |
| Arabian Sea | 2321 | 5.67 | 4.46 |
| Arabian Peninsula | 2021 | 4.94 | 4.58 |
| Northern Latitudes | 1919 | 4.69 | 0.36 |
| Tibetan Plateau | 628 | 1.53 | 1.27 |
| West Pacific | 582 | 1.42 | 0.14 |
| East China | 99 | 0.24 | 0.20 |
| Africa | 94 | 0.23 | 0.04 |

Figure 4 shows frequency distributions for both the average and maximum potential temperature of OTs (Figures 4a and 4b) and the average and maximum tropopause relative potential temperature (Figures 4c and 4d) over the study region for the entire study period. Here the average potential temperature of an OT refers to the average potential temperature of all the pixels that constitute that OT. The maximum potential temperature of an OT refers to the highest potential temperature reached by the pixels that constitute that OT. Tropopause relative values are calculated for each pixel using the local tropopause, before reporting the average and maximum of these values for all pixels within a given OT. As not all pixels of a given OT cross the tropopause, a small population of OTs have an average negative tropopause relative height. The color subdivisions indicate the proportions of cross-tropopause convection occurring during each month within each bin. As the full range of the distribution is broad, all OTs with potential temperature values outside of the plotted range are included in the lowermost and uppermost bins. The distributions of the high-value tails are shown in supplemental Figures 1 and 2.

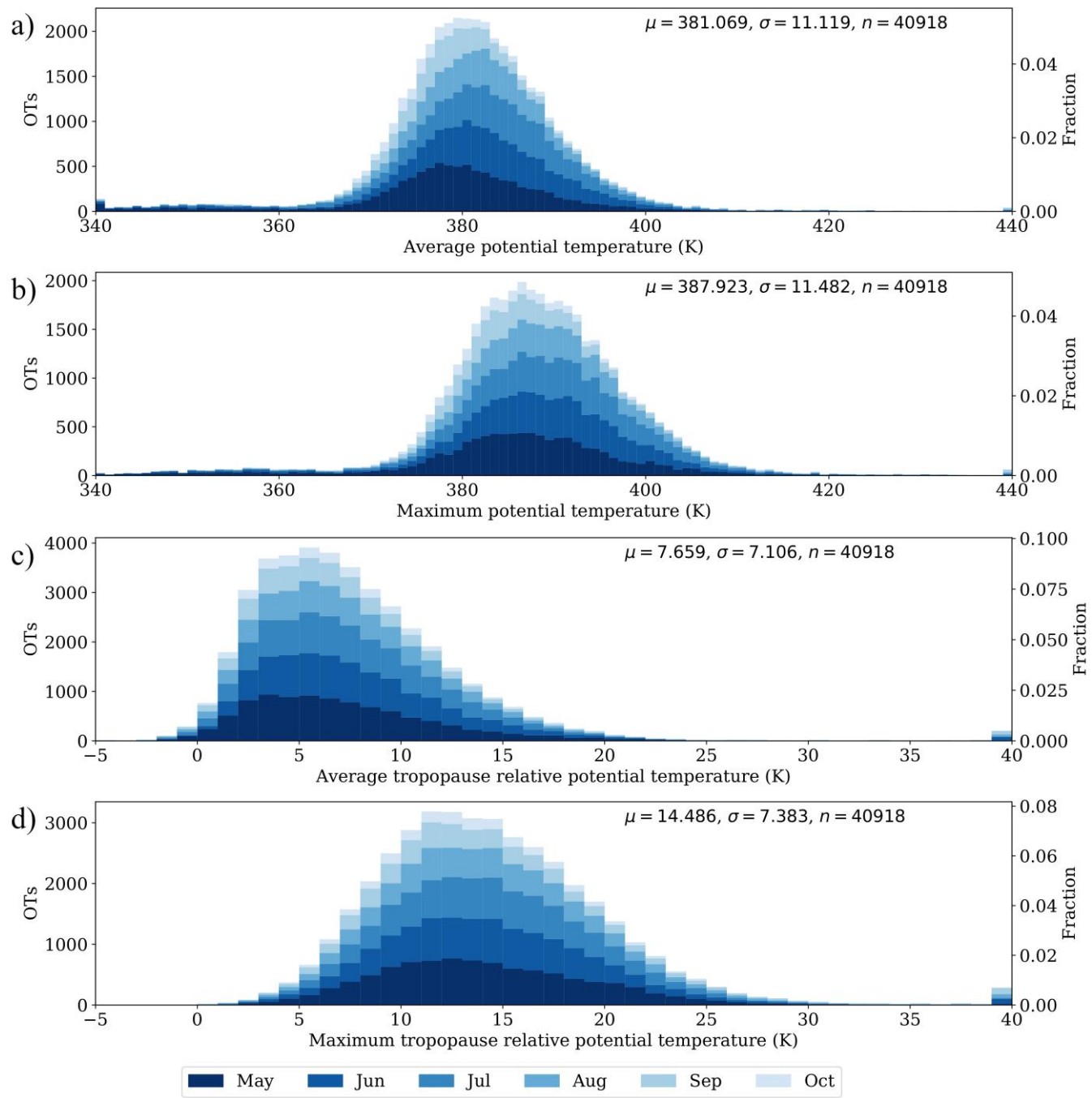

**Figure 4 The vertical distribution of all OTs within the study region. Figures 4a and 4b show the distributions of the average and maximum potential temperature of each OT binned by 1 K, respectively. Figures 4c and 4d show the distributions of the average and maximum tropopause relative potential temperature of each OT binned by 1 K, respectively. Within each bin, color indicates the fractional contribution of each month. For example, the 12 K average tropopause relative potential temperature bin is**

comprised of 315 May OTs, 344 June OTs, 376 July OTs, 227 August OTs, 141 September OTs, and 79 October OTs. The mean, standard deviation, and number of OTs for each distribution is indicated in the upper-right corner of each panel.

The average OT reached an average potential temperature of approximately 381 K and a maximum potential temperature of 388 K. In tropopause relative potential temperature this corresponds to +7.7 K and +14.5 K, respectively. In both potential temperature and tropopause relative potential temperature, the distributions shift higher from May through August, before descending again through September and October. This is most pronounced in the maximum potential temperature distribution, with the greatest number of OTs falling in the 386 K bin in May, rising to 389 K in August, before falling to 382 K in October. As this shift is present in tropopause relative potential temperature as well as potential temperature, it is indicative of convection reaching further into the stratosphere because the seasonal vertical motion of the tropopause is accounted for in the vertical coordinate. For further information on the tropopause during the study, supplemental Figure 3 shows the temporal evolution of the daily average of the local tropopause potential temperatures associated with each OT.

Both distributions exhibit long tails. This is seen in both directions in potential temperature space, with the lower values being most frequent in May. This is likely due to the climatologically lower tropopause earlier in the season. When normalized to the tropopause, however, the tail of the OT vertical distribution is largely at higher altitudes. This is visible in the large number of OTs within the highest bins (204 and 285 OTs for average and maximum tropopause relative potential temperature, respectively), which capture all OTs that reach a height above 40 K above the tropopause. These infrequent, extreme convective events account for approximately 0.6% of all OTs and occur in every month, although October has only one such OT (see supplemental Figures 1 and 2). Ultimately, 83.8% of the 40,918 OTs that occurred during the study period reached 380 K, with 9.7% reaching above 400 K, well into the lower stratosphere.

**3.2 Cross-tropopause convection source regions**

The intraseasonal distributions of cross-tropopause convection vary from region to region as shown in Figure 3a. In this section, we examine the individual distributions of the largest source regions.

Figure 5 shows the frequency distributions for both the average and maximum potential temperature of OTs over the South India region (Figures 5a and 5b) and over the North India region (Figures 5c and 5d). Cross-tropopause convection in the North India region is more numerous (11,844 OTs compared to 4,781 OTs) and reaches higher potential temperatures (392 K average maximum potential temperature compared to 389 K) than that over South India. This difference in potential temperature distribution, however, is primarily due to the latitude dependence of tropopause height. In tropopause relative coordinates, the average maximum potential temperatures are 392 K and 389 K for the North India region and South India region respectively.

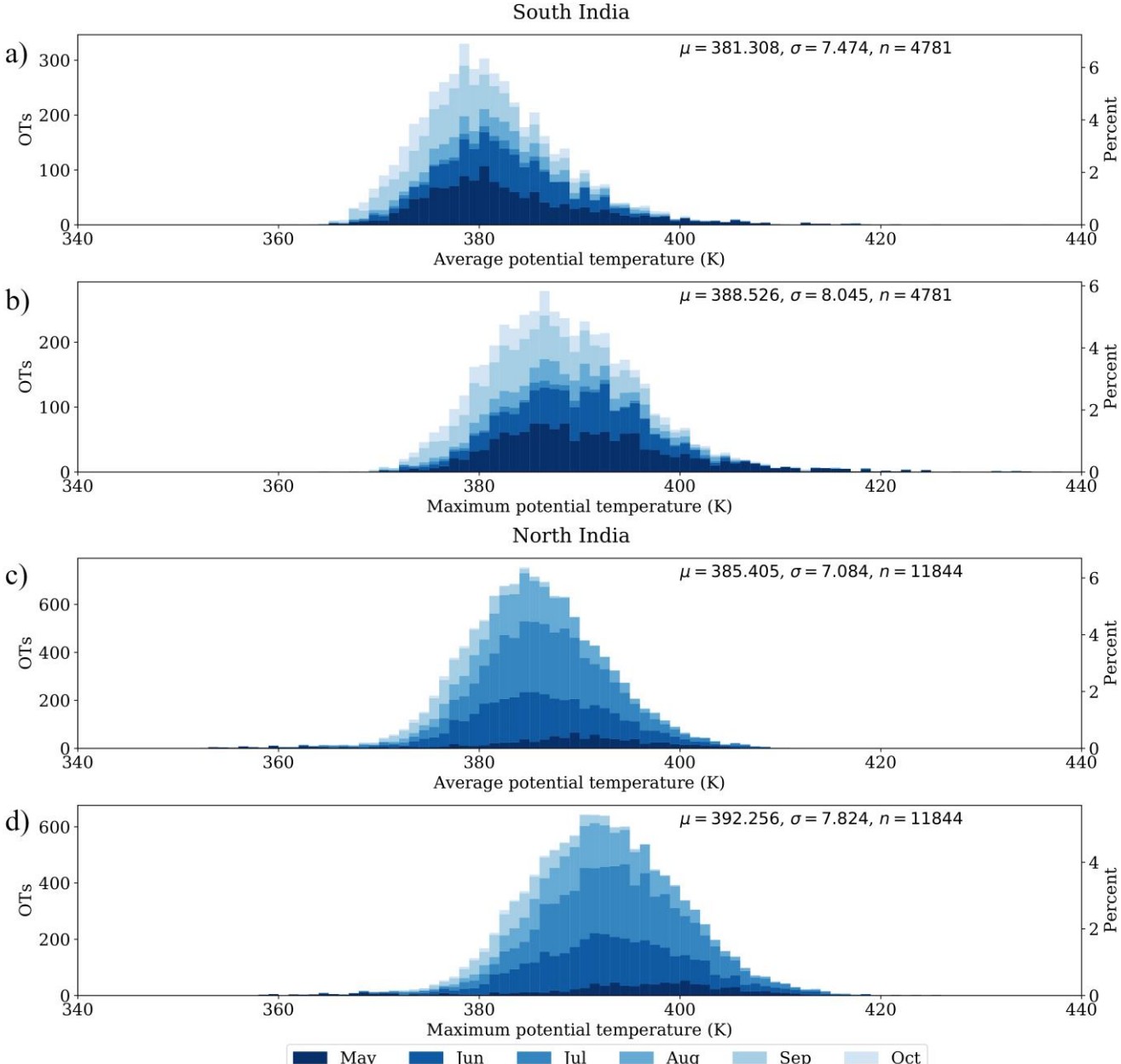

**Figure 5 The potential temperature distribution of OTs within the South India (Figures 5a and 5b) and North India (Figures 5c and 5d) regions. Figures 5a and 5b show the distributions of the average and maximum potential temperature of OTs occurring within the South India region binned by 1 K, respectively. Figures 5c and 5d show the distributions of the average and maximum potential temperature of OTs occurring within the North India region binned by 1 K, respectively. Within each bin, color indicates the fractional contribution of each month, as in Figure 4. The mean, standard deviation, and number of OTs for each distribution is indicated in the upper-right corner.**

The seasonality of the two regions also reveals a key difference. The South India region has the most cross-tropopause convection occurring in May (1380 OTs), June (1011 OTs), and September (1127 OTs), with few OTs appearing in July (152 OTs) and August (521 OTs). In contrast, over North India the majority of cross-tropopause convection occurs in July (4269 OTs), followed by June (3215 OTs) and August (2321 OTs), with relatively little in May (911 OTs) and September (1008 OTs). The distribution of cross-tropopause convection exhibits a northward shift from May through August, followed by a southward return in September.

This northward migration is consistent with the expected geographic evolution of the Asian monsoon (Abhik et al., 2013; Ganai et al., 2019; Goswami, 2011; Kajikawa et al., 2012; Romatschke et al., 2009; Sikka & Gadgil, 1980). Figure 6 shows the monthly geographic distribution of OTs (Figure 6, left column), average daily OLR (Figure 6, middle column), and average daily precipitation (Figure 6, right column). In both OLR and precipitation, a northward shift is visible in the geographic position of extreme values over land on the subcontinent from May through August, followed by a southward retreat in September and dissipation in October. In addition to the large-scale similarity in latitudinal intraseasonality, the regions of most frequent cross-tropopause convection are co-located with OLR and precipitation values indicative of significant convective activity. For example, the high density of OTs in northwest India during July corresponds to a region of low OLR and high precipitation. However, there are regions of low OLR and high precipitation that do not have corresponding OT activity, such as off the southwest coast of India during June. This is expected because cross-tropopause convection is a unique subset of extreme convection and not all convection, as represented by OLR or precipitation, is expected to result in OT activity, The agreement among multiple convective metrics suggests that the geographic and seasonal trends observed in OTs associated with cross-tropopause convection follow the development of the Asian monsoon.

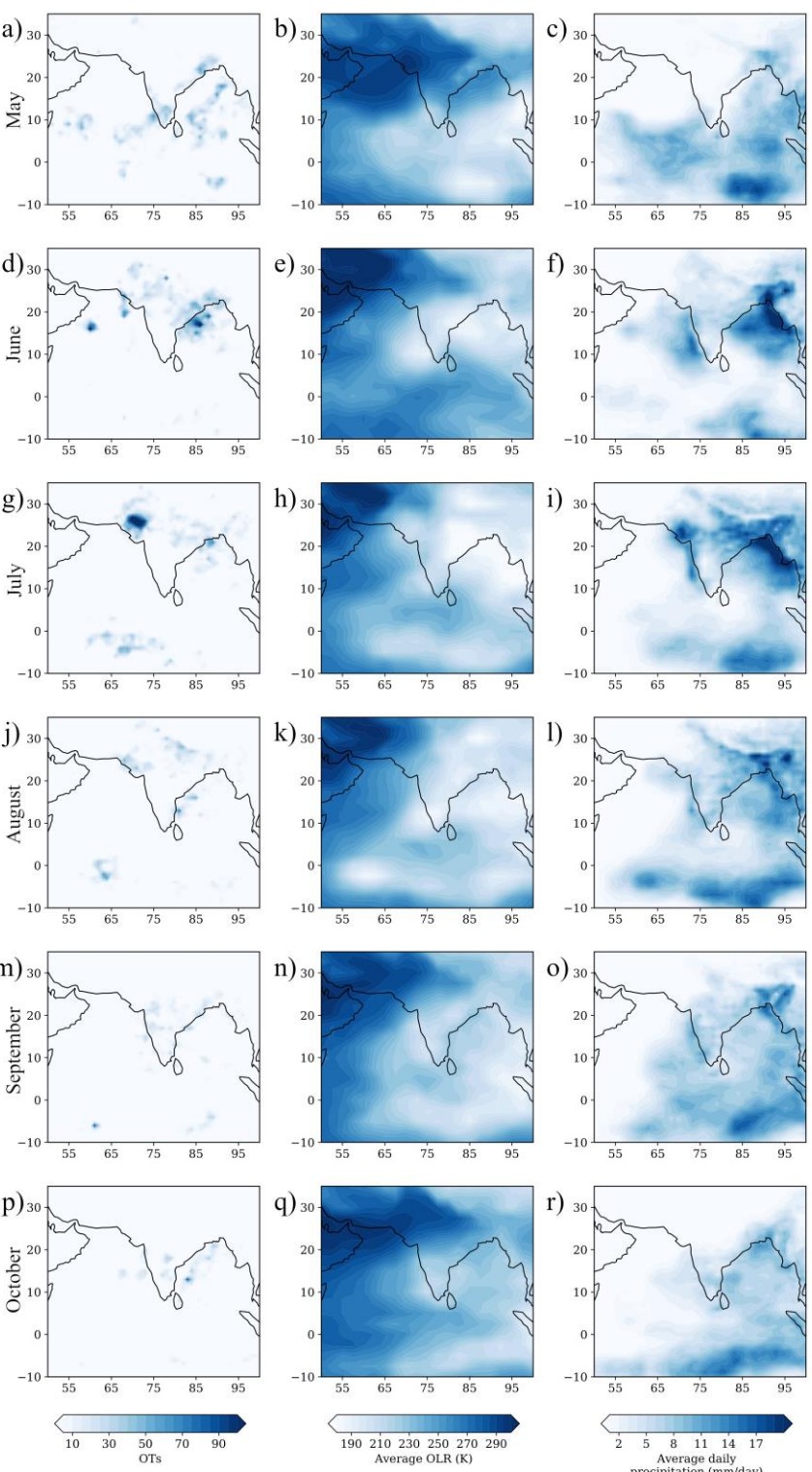

 **Figure 6 The geographic distribution of monthly convection as shown by OTs (left column), daily average OLR (middle column), and daily average precipitation (right column).**

Figure 7 shows the frequency distributions for both the average and maximum potential temperature of OTs over the Bay of Bengal region (Figures 7a and 7b) and over the Indian Ocean region (Figures 7c and 7d). Both regions have similar vertical distributions to the total OT distribution (Figure 4) in terms of the average (380 K and 386 K for the Bay of Bengal, and 381 K and 387 K for the Indian Ocean). The most significant contrast between these regions is the differences in the seasonality of cross-tropopause convection.

The Bay of Bengal has significant OT occurrence across all months, with the most activity occurring earlier in May and June (1574 and 1984 OTs, respectively). As shown in Figure 6, the monthly geographic distribution of the Bay of Bengal OTs matches the spatial distributions of low OLR and high precipitation. However, while the spatial distribution is replicated, the relative importance of each month in terms of number of OTs, is not distinguishable in either OLR or precipitation for the Bay of Bengal region. This is most apparent in the months of July and August (915 and 572 OTs, respectively), which exhibit the greatest areal extent of low OLR and high precipitation, but have fewer cross-tropopause convective events than May and June. This is likely due to the lower tropopause in the earlier months as shown in Figures 7a and 7b in which the lowermost potential temperature bins have a greater proportion of May and June contributions. Therefore, while convection over the Bay of Bengal may have a maximum in July and August, this convection is less likely to cross the tropopause.

The most notable feature of the distribution of cross-tropopause convection in the Indian Ocean region, is the relative lack of OTs in the month of June (482 OTs, 6.1%), compared to the other summer months with the next least convectively active month, August, having 1379 OTs (17.5%). Figure 6 suggests that this pause in cross-tropopause convection reflects a lack of convection more generally. Both the OLR and precipitation indices show that the Indian Ocean region has much less convective activity in June. The band of OTs clustered around the equator in July and August is consistent with the position of the southern band of the Indian Ocean's "double" ITCZ during the summer months (Berry & Reeder, 2014; Hu et al., 2007; Walsier & Gautier, 1993).

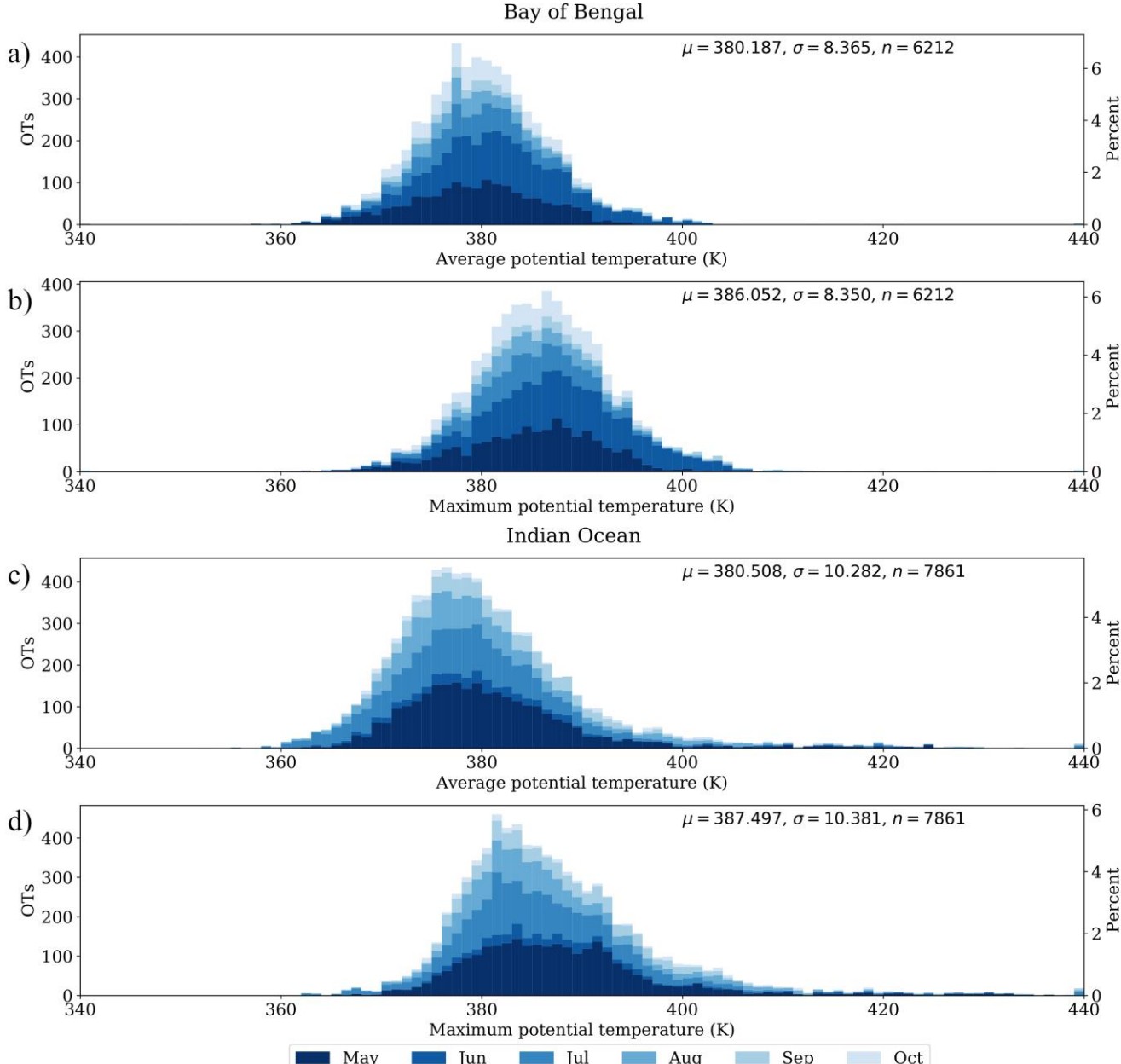

**Figure 7 As Figure 5, but for the potential temperature distributions of OTs over the Bay of Bengal region (Figures 7a and 7b) and the Indian Ocean region (Figures 7c and 7d).**

Figure 8 shows frequency distributions for both the average and maximum potential temperature of OTs over the Arabian Peninsula region (Figures 8a and 8b) and over the Arabian Sea region (Figures 8c and 8d). Though these regions are minority contributors to cross-tropopause convection over the entire study region (4.9 and 5.7% for the Arabian Peninsula and Arabian Sea regions, respectively), they exhibit high geographic concentrations of OTs comparable to the more

convectively active regions. The Arabian Peninsula and Arabian Sea regions have maximum OT densities of 174 and 138 OTs per 1°x1° grid cell, respectively, compared to 154 OTs per 1°x1° grid cell for the North India region. Further, while the vertical distribution of the cross-tropopause convection in these regions is similar to the overall distribution across the entire study region, they have distinct seasonal features.

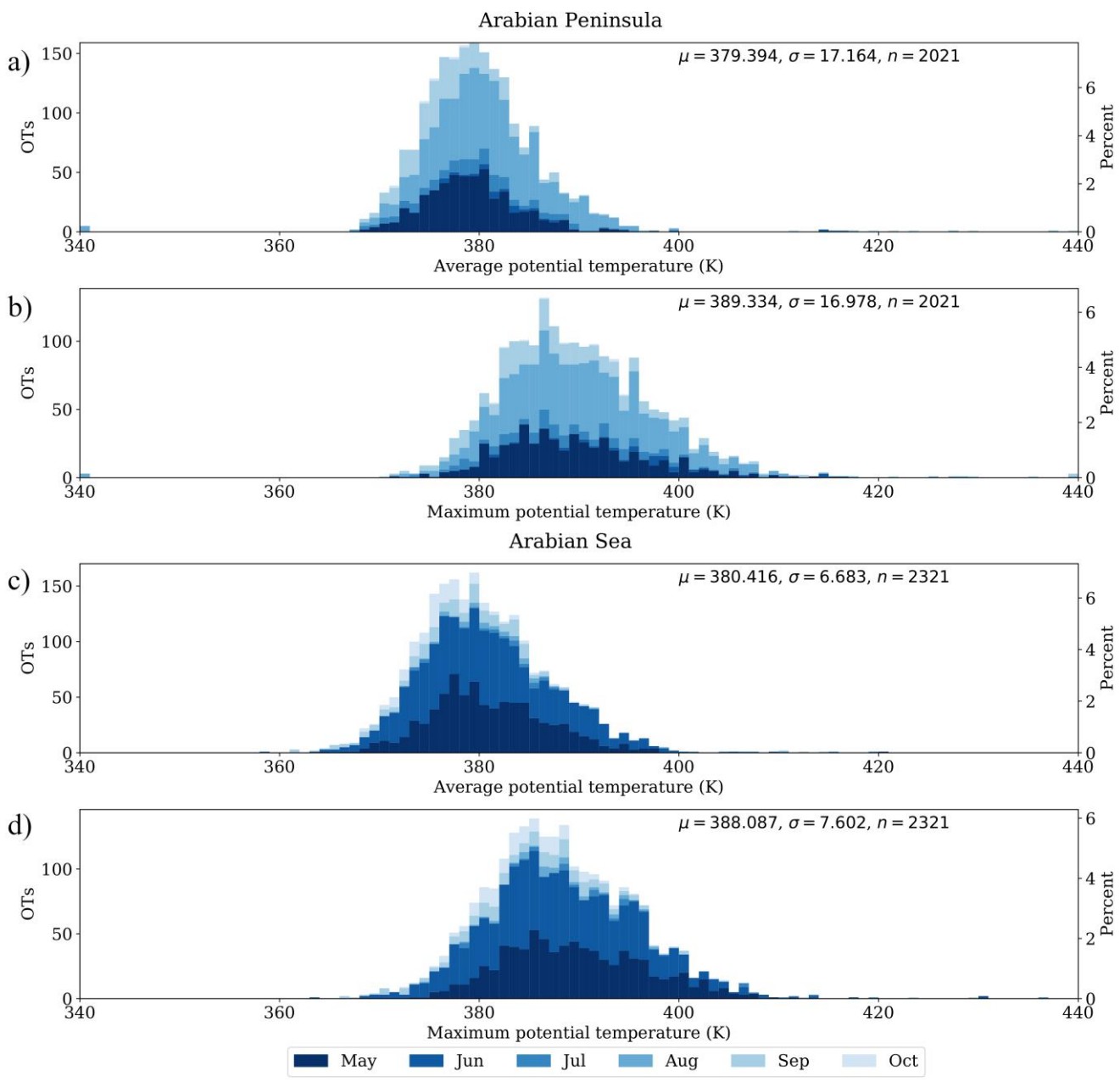

Figure 8 As Figure 5, but for the potential temperature distributions of OTs over the Arabian Peninsula region (Figures 8a and 8b) and the Arabian Sea region (Figures 8c and 8d).

The cross-tropopause convection in the Arabian Peninsula region occurs almost exclusively in May and August (which together contribute 1459 OTs, 72.2%). June, July and September only contribute 41 (2.0%), 164 (8.1%), and 347 OTs (17.2%) respectively. The OTs in this region are co-located with the Asir Mountains, which run parallel to the Red Sea coast suggesting that the cross-tropopause convection is likely orographic in origin (Abdullah & Al-Mazroui, 1998; Chakraborty et al., 2006; Segele & Lamb, 2005). Figure 9 shows the monthly geographic distribution of OTs (Figure 9, first column), average daily OLR (Figure 9, second column), average daily precipitation (Figure 9, third column), and average daily specific humidity at 850 hPa (Figure 9, fourth column). Although OLR and precipitation show similar local maxima in the same region of the southeast coast of the Red Sea, they do not reproduce the relative intensity evident in the OTs. The specific humidity at 850 hPa, however, shows more intense local maxima along this coastline (Figure 9p), further indicating orographic lifting of this moist air as a likely source of cross-tropopause convection in this region.

The OT distribution over the Red Sea region is in part decoupled from larger scale patterns evident in OLR and precipitation. The different distributions agree in that the most convectively intense month according to the OLR and precipitation distribution occurs in August. However, both OLR and precipitation indicate more intense convection is occurring over Ethiopia than over the southeast coast of the Red Sea, and that there is substantial convection during June and July, when there is little evidence for cross-tropopause convection in the OTs. This indicates that the southeast coast of the Red Sea may be uniquely conducive to cross-tropopause convection, possibly due to orographic initiation. As the study region does not extend further into Africa and the study period is limited to 2017, this conclusion is preliminary. Nonetheless, global studies of extreme convection also indicate more convective activity over the Arabian Peninsula than over Ethiopia (Liu & Lie, 2016; Liu et al., 2020). Further research into the circumstances that result in this tight geographic concentration of cross-tropopause convection is warranted.

The Arabian Sea region has most of its cross-tropopause convection confined to the months of May and June as seen in Figures 8c and 8d. During August-October combined, only 404 OTs (17.4%) occur. The seasonal dependence of OTs is also reflected in other measures of convection in this region. As before, Figure 6 shows the monthly geographic distribution of OTs (Figure 6, left column), average daily OLR (Figure 6, middle column), and average daily precipitation (Figure 6, right column) of this region. Precipitation in particular, has maxima co-located with the regions of most OT activity during May and June. OLR has a similar geographic distribution, but does not resolve the fine scale variability across the region. For example, the "point" maxima in June is present in the OLR distribution, but is not as intense relative to the overall region. The June cross-tropopause convection is so geographically concentrated that it is likely sourced from a single large storm system that passed through the Arabian Sea during the first week of June. The region is in its tropical cyclone season at this time (Deshpande et al., 2021; Evan et al., 2011). Supplementary Figure 4 shows the passage of the large storm system that is the likely source of the cross-tropopause convection in the Arabian Sea in visible reflectance near 0.6 microns data from the gridded International Satellite Cloud Climatology Project B1 data (Knapp et al., 2011). We note that this geographic "point" source of convective influence over the Arabian Sea is also visible in a study by Legras and Bucci (2020), of the 2017 Asian monsoon.

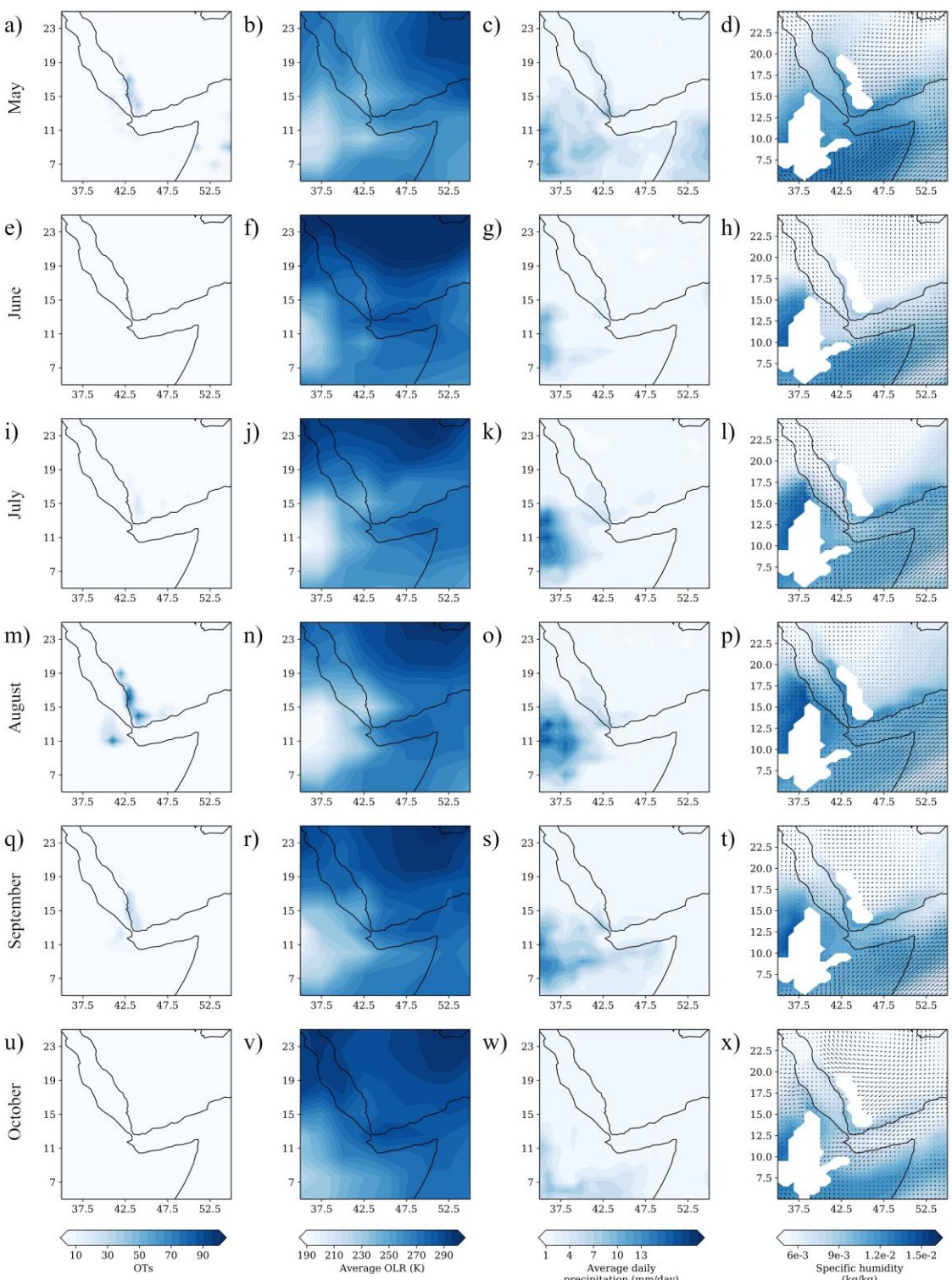

**Figure 9 The geographic distribution of monthly convection over east Africa and the southwest Arabian Peninsula as shown by OTs (first column), daily average OLR (second column), and daily average precipitation (third column). Additionally, the specific humidity and wind vectors at 850 hPa from MERRA-2 are shown (fourth column).**

## 4 Summary and discussion

We construct a database of cross-tropopause convection in the Asian monsoon region for the months of May through October of 2017 using Meteosat-8 geostationary satellite detections of OTs. We analyze 40,918 individual OTs to identify geographic regions of high frequency and high density cross-tropopause convection and to characterize their vertical distributions and intraseasonal variability. Additionally, through comparison with OLR and precipitation observations we place the cross-tropopause database in the context of tropospheric convection generally.

We find that cross-tropopause convection is active in the Asian monsoon region during the months of May through August (with daily averages of these months above 300 OTs/day) and declines through September and October. Most of the OTs occur over South Asia with North India contributing 29.0% of all OTs, South India contributing 11.7% of all OTs, and the Bay of Bengal contributing 15.2% of all OTs. Together with the Indian Ocean region (19.2%), the most cross-tropopause convection occurs in these regions, and they cumulatively account for 75.1% of all OTs. While the Arabian Peninsula and Arabian Sea regions are smaller sources of cross-tropopause convection (4.9% and 5.7%, respectively), they exhibit geographic OT densities comparable to the major source regions. The majority of cross-tropopause convection within the entire study region reaches a maximum height above 380 K (83.8%), with an average OT maximum height of 387 K corresponding to 14.5 K above the tropopause.

We further identify distinct intraseasonal behavior within these subregions. The four regions of most cross-tropopause convection (North India, Indian Ocean, Bay of Bengal, and South India) correspond to the areas of significant general tropospheric convection identified by the average OLR and precipitation. Within these regions, the geographic distribution of cross-tropopause convection most closely matches the distribution of intense convection as shown by the minimum OLR and maximum daily precipitation. For the North India, South India, and Bay of Bengal regions this results in the distribution of OTs following the north-south movement of the development of the Asian Monsoon across the study period. These findings are largely in agreement with Vogel et al. (2015), which examined the intraseasonal variability of convective source regions within the Asian monsoon region during 2012 with small differences likely arising from differences in time period considered and in the spatial regions considered.

Within the Arabian Peninsula and Arabian Sea regions, the seasonal and geographic distributions are controlled by the sources of cross-tropopause convection unique to these regions. In the Arabian Peninsula, OTs are primarily observed in May and August, corresponding to a confluence of low-level moisture along the western slope of the Asir Mountains indicating a likely orographic source of convection. In the Arabian Sea, most OTs occur in June, with a "hotspot" corresponding to a single large storm system. Whether these behaviors in cross-tropopause convection observed in 2017 are recurring features should be explored in future research.

Our examination of cross-tropopause convection is complementary to the large body of research that has examined the effects of deep convection that reaches the upper troposphere and subsequently enters the lower stratosphere through diabatic ascent. There has been disagreement in prior work regarding whether oceanic or land-based convective source regions for the Asian monsoon anticyclone are more important. For example, many studies have found that convection over the Tibetan Plateau is the primary contributor (Bergman et al., 2013; Fu et al., 2006; Heath & Fuelberg, 2014; Pan et al., 2016; Wu et al., 2020). In contrast, other studies have shown that the Bay of Bengal or the west Pacific are the majority contributors (Chen et al., 2012; Devasthale & Fueglistaler, 2010; James et al., 2008). Here, differences between the reanalyses used in modelling studies of Asian monsoon anticyclone convective source regions (e.g. Bergman et al., 2013; and Chen et al., 2012) contribute to the discrepancies amongst their conclusions. We find that both oceanic and land-based regions contribute significant amounts, though with different seasonal distributions.

Further, the highest frequency of land-based cross-tropopause convection is centered on North India, not the Tibetan Plateau. This is a significant difference from studies that find the Tibetan Plateau to be the primary conduit by which tropospheric convection reaches the lower stratosphere (Bergman et al., 2013; Pan et al., 2016). Our analysis of cross-tropopause convection most agrees with the studies that conclude that the Tibetan Plateau is a region favorable to diabatic ascent but not itself a major convective source (Legras & Bucci, 2020; Tissier & Legras., 2016). Moreover, Tissier & Legras (2016) also show both oceanic and land-based cross-tropopause convective sources within the Asian monsoon region in agreement with our spatial distribution.

Within the context of the StratoClim project, our results indicate that cross-tropopause convection is not uncommon, in agreement with the in-situ observations of elevated water vapor in the LS attributed to cross-tropopause convection (Khaykin et al., 2022; Lee et al., 2019). We also find that the attribution of convective influence in the LS to land-based convection over India during the StratoClim research flights (Bucci et al., 2020; Khaykin et al., 2022) matches the most convectively active region, as indicated by OT distribution, during that time period of late July and early August.

As this analysis only covers May through October of 2017, it is unable to assess any interannual variability that may alter the seasonal and geographic characteristics identified here. Furthermore, the geographic boundaries of the OT dataset from the Meteosat coverage do not include two areas that merit further investigation. First, is the convective "hotspot" in the western Pacific Ocean, at approximately 135° E, that has been identified as a potential key region for convective influence on the UTLS (e.g. Chen et al., 2012). Second, along the west boundary of our study region, there is indication of additional cross-tropopause convective over Africa, and in particular the African Monsoon region may be an area of significant OT activity. Expanding this research to analyze additional years would allow for an assessment of interannual variability and improve the identification of region specific properties. Further, additional work to normalize geostationary satellite observations across platforms would facilitate assessments of the global impact of cross-tropopause convection.

Overall, this analysis demonstrates the importance of cross-tropopause convection when considering the influence of convection on the lower stratosphere within the Asian Monsoon region. Not only does the time-scale of the transport mechanism differ from the diabatic ascent pathway, resulting in potentially different chemical species reaching the lower

stratosphere, the geographic distribution of cross-tropopause convection indicates a need to account for regions of convective activity beyond those previously identified as primary sources of convective influence via diabatic ascent (e.g. northwest India in Figure 2). Further, the distribution of cross-tropopause convection shows that both oceanic and land-based regions must be considered for a complete accounting of convective influence on the Asian monsoon anticyclone. Our work also demonstrates that it is critical to consider multi-month time-scales given the large intraseasonal differences in the contributions of each source region in agreement with Vogel et al. (2015). Confining an analysis to a single month would over-emphasize certain source regions while missing others. For example, considering only August would show that North India has more convective activity than the Bay of Bengal, though the inverse is apparent in May. Given the importance of timing and location when assessing the potential transport of pollution into the lower stratosphere in the Asian Monsoon region, in addition to its importance in determining subsequent transport pathways into the global stratosphere, a perspective on convective impacts on the lower stratosphere that includes cross-tropopause convection and covers the entire spatiotemporal region of convective activity is critical.

**Acknowledgements**

We would like to thank the members of the Anderson group for their support. This work has been supported by the National Aeronautics and Space Administration (NASA) under NASA award NNX15AF60G (UV Absorption Cross Sections and Equilibrium Constant of ClOOCl Determined from New Laboratory Spectroscopy Studies of 18 ClOOCl and ClO) and a grant from the National Science Foundation (NSF) Arctic Observing Network (AON) Program under NSF award 1203583 (Collaborative Research: Multi-Regional Scale Aircraft Observations of Methane and Carbon Dioxide Isotopic Fluxes in the Arctic).

**Competing Interests**

The authors declare that they have no competing interests.

**Author Contributions**

CEC designed the study and performed the analysis. KMB developed the OT detection algorithm and applied it to the Asian monsoon region. JBS provided scientific analysis and assisted with figure development. JGA supervised the study. CEC wrote the manuscript; all co-authors edited and revised the manuscript.

**Data availability**

Meteosat-8 multispectral imagery: https://navigator.eumetsat.int/start

MERRA-2 reanalysis data: https://disc.gsfc.nasa.gov/

NOAA OLR data: https://psl.noaa.gov/data/gridded/data.olrcdr.interp.html

GPCP Climate Data Record v. 1.3: https://rda.ucar.edu/datasets/ds728.7/

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
