# Peer review of "Distribution of cross-tropopause convection within the Asian monsoon region from May through October 2017"

_Atmospheric Chemistry and Physics, 2022_

## Referee Comment (RC2)

**Review of C. E. Clapp et al. "Distribution of cross-tropopause convection within the Asian monsoon region from May through October 2017"**

The manuscript by Corey Clapp and coauthors provides an analysis of geographic and intraseasonal distribution of cross-tropopause convection in the Asian Summer Monsoon region. The analysis relies on the overshooting tops database spanning a full monsoon season in 2017 and including nearly 41,000 events derived from Meteosat-8 geostationary imagery data. The distribution of overshooting tops is compared with OLR and precipitation data. The study points out significant intraseasonal and spatial variability of tropopause-overshooting convection in the Asian monsoon region.

The study represents a valuable contribution to a better understanding of the source regions and variability of the tropopause-overshooting convection in the Asian monsoon region. With that, the presentation of the results and their current context require careful revision before acceptance to ACP.

General remarks.

- The 2017 Asian monsoon season addressed by the study was the target of StratoClim campaign, which included extensive airborne and balloon-borne measurements across the Southern slopes and the North India regions. There is a large number of studies that followed this experiment, some of which are cited in the paper however this overview is far from being complete. I suggest that the authors make sure to mention all the relevant papers and carefully discuss the present results in relation with the previous work. More specific remarks on that matter are provided hereinafter.

- The graphical material is often hard to read, this is particularly the case for the OT maps. I suggest to use a different color map or otherwise make sure that the coastlines are clearly visible in each panel.

- I wonder if is the authors could provide an estimate of the total OT area, which would help understanding the magnitude of the impact of tropopause-overshooting convection. Such information could be used to constrain the modeling studies.

- Given the content of the last section, it should rather be named "summary and discussion"

Specific remarks.

L29 – 37. Here the authors define the study objectives. I would suggest to move it towards the end of the Introduction.

L40 – 45. The referencing should be completed with StratoClim studies, such as Brunamonti et al., ACP 2018; Lee et al., 2021; Lamraoui et al., ACPD 2022). In particular, the source regions for the cross-tropopause convection are discussed in detail by Khaykin et al., ACP 2022.

L55 – 57. For the effects of eddy shedding one might refer to Fujiwara et al., ACP 2021. For the transport of Asian pollution towards midlatitudes a pertinent reference would be Khaykin et al., ACP, 2014

Fig. 1 caption. The description of the panels should be in order

L.123. Fig 1c -> Fig. 1d ?

L. 183-185. I am not sure to understand the line of logic here. What is meant by the particular efficiency of TB due to central location?

L.191-199. This paragraph is particularly difficult to follow. The term "convective activity" seems to be used for both the OT and OLR features, which renders unclear the discussion on their similarities.

L.202 I do now see any significant agreement between OT and precipitation distributions. Overall, I find the discussion that follows largely unclear.

Fig. 2. The panels are too small and barely readable. It is nearly impossible to distinguish between the different curves of similar color.

Fig.2 caption. Wrong referencing to Fig. 1

L.220-225 I believe a brief mention on the limited representativeness of OT evolution would be pertinent here.

Fig. 3 Where does an isolated feature at highest altitude come from?

L. 244. Altitude distribution -> vertical distribution

L.254 redundant with Fig. 3 caption.

L.396 I do not fully agree with the statement regarding the match between OLR, precipitation and OT, or at least it is not obvious from the figures. Alternatively, if that is indeed the case, does the OT analysis provide an added value for a better characterization of ASM convection?

---

## Author Response (AR1)

**Response to Referee 1**

We thank the reviewer for their insightful and thorough criticism. By addressing their comments, we have improved the study as described below. The reviewer's comments are shown in black, and our responses are shown in blue.

This is an interesting and important study, which merits its publication in ACP. However the scientific content, the quality of the study and its presentation (in particular the figures) should be further improved. I suggest some major revisions before publication by ACP.

General comments:

1) The quality, the size and the description of the figures should be improved. Some of the figures should be enlarged and it would be helpful to get a more detailed description of the presented quantities (more details see below).

The presentation of the figures, including the size and description, has been improved as recommended. In general, the figures were reorganized to allow for significant enlargement and text has been added to provide additional details to the descriptions. The specifics of the changes to each figure will be discussed in our responses to the comments below.

2) In the last years, many publications are published regarding convection in the region of the Asian monsoon. The StratoClim aircraft campaign was performed in Kathmandu (Nepal) in summer 2017 probing air in the Asian monsoon anticyclone conducted in same year as the study by Clapp et al, ACPD, 2022. I think it is worth to discuss some of the results of the StratoClim aircraft campaign in connection to the results by Clapp et al, ACPD,2022 to demonstrate differences and similarities found for 2017 (some of the papers related to StratoClim aircraft campaign are already mentioned by Clapp et al, ACPD,2022 e.g. Legras & Bucci, 2020; Johannson et al, 2020, von Hobe et al., 2021...). There is a special issue to StratoClim in ACP: https://acp.copernicus.org/articles/special_issue1012.html. Here (and also in the references), the authors can find more literature related to convection and vertical transport in the region of the Asian monsoon. I recommend to revise the introduction a bit to give the reader a more comprehensive overview about the topic.

A specific discussion of the results of the StratoClim campaign has been added to the introduction (lines 101-114) to provide additional context to our study, particularly an emphasis on the selection of 2017 as the year of study for the sake of comparison (Brunamonti et al., 2018; Bucci et al., 2020; Johansson et al., 2020; Khaykin et al., 2022; Lee et al., 2019; Lee et al., 2021; Legras & Bucci, 2020; Nützel et al., 2019; Vogel et al., 2019; von Hobe et al., 2021; Yan et al., 2019). This discussion is also extended to a section in the conclusion (lines 493-506).

3) It is sometimes difficult to follow the presented analysis. Some more detailed description and motivation would be helpfully for a better understanding (details see below). In particular the wording is sometimes strange e.g.' We identify seasonal trends'. I think it should be called intraseasonal variability or evolution. A trend would cover a much longer time period (e.g. an increasing trend for OTs occurrence during the last ten years in the Asian monsoon region). I recommend to change the wording throughout the manuscript. 4) The tropopause height is a key

parameter for the analysis presented. More details about its calculation and variability should be presented (more details see below).

We agree that the verbiage of 'trends' was vague and improperly applied. The wording throughout the manuscript has been updated to 'intraseasonal variability' as recommended. Additional details about the usage of tropopause height has been added to the Results section (lines 298-304). A supplementary section including a figure showing the distribution of local tropopause heights of the OT database has also been added (supplementary Figure S3). Our responses to specific concerns about additional details are provided below.
* * *
Major comments:

P1 L22: 'the contributions across the entire region'. Unclear, please clarify this.

The sentence (lines 21-24) was edited as follows for clarification: "This work demonstrates that when evaluating the effects of convection on lower stratospheric composition over the Asian monsoon region it is important to consider the impact of cross-tropopause convection specifically, as well as the contributions from both land-based and oceanic regions due to the significant geographic and monthly variation in convective activity."

P1/2 L25-40: Convection in the Asian monsoon region is discussed for many years. Therefore several references regarding convection in the Asian monsoon region exists up to now. Please add here some more relevant citations.

Additional pertinent citations addressing the role of convection in the Asian monsoon region have been added to relevant sentences within the paragraph (lines 26-39) referred to here. (Brioude et al., 2010; Claxton et al., 2019; Lelieveld et al., 2018; Park et al., 2007; Pisso et al., 2010; Randel et al., 2010; Santee et al., 2017; Tegtmeier et al., 2020).

P2 L41: 'Satellite observations have shown consistent indications of tropospheric influence in water vapor and carbon monoxide maxima as well as ozone minima concurrent with the Asian monsoon anticyclone that develops in the UTLS region'. Enhanced mixing ratios of more chemical trace gases tropospheric-origins than $H_2O$ and CO as well as of aerosol are found within the Asian monsoon anticyclone compared to the background air in these altitudes. Please revise this sentence and be a bit more comprehensive. This is mentioned later in L46-53. That is confusing, please rearrange these parts of the introduction.

The sentence has been reworded and the paragraph (lines 40-56) has been reorganized to introduce the compositional characteristics of the Asian monsoon anticyclone more comprehensively in conjunction with the following paragraph.

P2 L44: What means 'these perturbations'. Please clarify.

Additional clarifying text (lines 48-49) has been added: "the convective source of these water vapor, carbon monoxide, and ozone perturbations."

P2 L51: Aerosol measurement in the region of the Asian monsoon is discussed in several publications. It would be fair to add in addition to Vernier et al. some other studies e.g. by Höpfner et al., Nat. Geosci., 2019 (https://www.nature.com/articles/s41561-019-0385-8); Hanumanthu et al., ACP, 2019 (https://doi.org/10.1029/2003JD003770); Brunamonti et al., 2918 (https://doi.org/10.5194/acp-18-15937-2018); Bian et al., Natl. Sci. Rev., 2020 (https://doi.org/10.1093/nsr/nwaa005).There exist even more paper. The authors should decide which references are most suited.

Additional suitable references, including those recommended, have been added (lines 52-56). (Bian et al., 2020; Brunamonti et al, 2018; Hanumathu et al., 2020; Höpfner et al., 2019).

P2 L53: Enhanced VSLS were measured within the Asian monsoon anticyclone in 2017 and their ODPs were calculated (see Adcock et al., JGR, 2021; https://doi.org/https://doi.org/10.1029/2020JD033137)

A citation of Adcock et al., 2021 was added (line 55).

P2 L54: The authors should add a short explanation of the general upward transport in the region of the Asian monsoon.

A short paragraph (lines 57-65) detailing the general upward transport in the Asian monsoon region with relevant citations was added to the introduction.

P2 L61: 'The location and timing of the initial deep convection influence both the chemical impact of convective transport on the composition of the UTLS and the geographic distribution of that impact.'--> This sounds odd. Maybe better 'The location and timing of the initial deep convection influence the kind and amount of tropospheric source gases and aerosol transported into the altitudes of the Asian monsoon anticyclone (or in UTLS altitudes) .....'

The sentence (lines 73-74) was reworded as follows: "The location and timing of deep convection influence the kind and amount of tropospheric source gases and aerosol transported into the AMA."

P2 L64: 'therefore which regions are impacted "downstream" in the large-scale circulation.' Please clarify.

The phrasing (lines 76-77) has been clarified as follows: "Furthermore, the location of the initial convective transport determines the pathway of entry into the lower stratosphere and therefore which regions are impacted by subsequent transport within the large-scale circulation."

P3 L70-79: Many of the cited studies are base on model simulations. In models, different treatments of convection are used (e.g different meteorological reanalyses, different parameterisations of convection) yielding different results. I think this should be mentioned somehow. Further, it would be useful to distinguish in the discussion between studies that are based on model simulations from studies that are based on observations to identify the locations of convective sources.

In this paragraph (lines 82-94) additional text noting which studies are based on model simulations, and which are based on observations has been added. The potential effects of different treatments of convection within models were also noted.

P4 L117: 'analyzing large scale trends' Please specify, trends of what? (see general comment 3.)

As recommended in comment 3, the phrase (lines 156-159) has been reworded in terms of intraseasonal variability: "The OT database, however, is still valid for analyzing intraseasonal variability, evolution, and distribution because the consistent and frequent sampling, the long time period under study, and the large region of interest retain the major features of OT frequency, depth, and geographic distribution."

P4 L123: '(see Figure 1c)' -> '(see Figure 1d)' In the text, first Fig. 1a should be cited. Please rearrange the figures accordingly. I recommend to first show Fig. 1d as a separate Fig. 1 (+corresponding legend, details see below).

Figure 1 has been rearranged such that Fig. 1d is now a separate figure that is introduced prior and is referenced first. As a result, the original Figure 1 is now Figure 2.

P4 L124: 'to allow for comparison to prior work' Please add some references to prior work.

Additional references were added to this sentence (lines 164-165): "e.g. Bergman et al., 2013; Fu et al., 2006; Heath & Fuelberg, 2014."

P5 L130: Please add some more recent references.

More recent references have been added to lines 167-168 (Bhat & Kumar, 2015; Saikranthi et al., 2018; Virts & Houze, 2016).

P5 L139: 'The Indian Ocean region captures and separates the influence of the ITCZ, seen in the OT distribution (Figure 1a), from the other two regions.' Sentence in unclear, please specify which other two regions. For me it looks like that the Indian subcontinent is located between the regions with highest OTs occurrence. Add a legend for the regions to Fig. 1d to avoid any misunderstanding. I recommend to show Fig. 1d (+ Legend) as separate Figure 1.

The sentence (lines 182-184) was reworded to specify the other two regions: "The Indian Ocean region captures and separates the influence of the intertropical convergence zone (ITCZ), seen in the OT distribution (Figure 2a), from the other two oceanic regions (Bay of Bengal and Arabian Sea)." As recommended, Fig. 1d has been separated into an independent Figure 1 with clear labelling of the regions.

P5 L140: Convection on the northwestern Indian subcontinent was discussed i.a. in Höpfner et al., Nature Geoscience, 2019, as a potential source for the Asian tropopause aerosol layer (ATAL) in summer 2017. I think it is worth to mentioned it here. Further Khaykin et al., ACP, 20022 (https://doi.org/10.5194/acp-22-3169-2022) identified several convective source contribution to air masses probed during the StratoClim aircraft campaign in 2017.

The Höpfner et al. (2019) reference was added here (line 181), and the Khaykin et al. (2022) reference was integrated into several sentences (e.g. lines 71, 108, and 495) where it was pertinent.

P6 L160: Fig. 1a should be enlarged (e.g show only the dotted-white box). It is good to show the entire Asian monsoon anticylone, however the main regions of OTs occurrence are difficult to see. Maybe it would be better to focus here on the OTs and show the anticylone in Fig.1b/c. Further, something is wrong with the labels at the y-axis.

Figure 1 (now Figure 2, see above comment) has been substantially revised, including enlarging Fig. 2a, making the depiction of the anticyclone Montgomery contours less intrusive, improving the general visibility, and the separation of the previous Fig. 1d.

P6 L163: 'The Indian Ocean shows a high volume of dispersed cross-tropopause convection located at the ITCZ in an east-west band between 0-5° N. ' This sentence is confusing. During boreal summer the ITCZ is located further north of 0-5° N.

The sentence (lines 209-210) was rephased for clarity given the general understanding of the ITCZ: "The Indian Ocean shows a high volume of dispersed cross-tropopause convection located in an east-west band around the equator." The Indian Ocean, however, has a "double" ITCZ during the summer months (Berry & Reeder, 2014; Hu et al., 2007; Waliser & Gautier, 1993) with a southern band that has overlap with the OT cluster. Discussion of the overlap was added to the section dedicated to the OT distribution within the Indian Ocean region (lines 394-400).

P6 L167: 'This high count of intense convective events has been previously observed' -> 'This high count of intense convective events over the Arabian Peninsula ...'

The suggested clearer wording was implemented (lines 257-258).

P6 L167: 'The distribution shows..' -> 'The spacial distribution of OTs over Asia (Fig. 1a) shows ...

The suggested clearer wording was implemented (lines 213-215).

P/ L181: I think is worth to mention, that Legras and Bucci (2020) is also related to the Asian summer monsoon 2017 just as Clapp et al., ACPD, 2022. P/ L184 '..but the subsequent diabatic ascent into the lower stratosphere occurs primarily in neighboring regions'. Please explain this general upward transport in the region of the Asian monsoon in more detail.

A discussion of the general upward transport in the Asian monsoon region was added to the introduction (lines 57-65) and recalled here (lines 225-229). The overlap in study time periods between this study and Legras and Bucci (2020) was also noted (line 231).

P7 L191: 'Figures 1b and 1c show the OLR minimum and average daily value for the entire study period.' as mean values over the study period from 1 May to 31 October 2017. Yes or trend ?? Please clarify.

The confusing "trend" wording was replaced with "distribution:" "The minimum OLR spatial distribution, as a proxy for the "deepest" convection, matches the cross-tropopause convective distribution better than the average OLR." (lines 238-240)

P7 L200: 'Figures 1c and 1d show the GPCP maximum daily precipitation and average daily precipitation for the entire study period.' in each grid-box? 'Figures 1c and 1d' --> 'Figures 1e and 1f'

The figures show the GPCP maximum daily precipitation and average daily precipitation for the entire study period for each grid-box. Text indicating this was added (lines 248-249). The figure misattribution was also corrected.

P8 Fig. 2: caption 'Figure 1' -> 'Figure 2'. Please enlarge the height of the panels. It is difficult to see the different lines.

The caption was fixed (beginning at line 288), and the figure (Figure 3) reorganized with the months as rows rather than columns to allow for a significant enlargement of all the panels.

P8 L222: 'For example, the active months of May through August contribute on average 22.9% of the total OTs while September and October contribute 13.4% and 6.3%, respectively.' I can not see these percentages in Fig2. Please explain this in more detail.

The sentence was split in two (lines 262-264) and detail was added to clarify that the percentages cited reflect the cumulative number of OTs that occurred within each month and were not the average daily values shown in the figure: "For example, during the active months of May through August, on average 22.9% of the total OTs occur per month. In contrast, during the months of September and October 13.4% and 6.3% of total OTs occur, respectively."

P8 L231: I would not call this 'outlier'. Better something like 'OT caused by specific meteorological conditions'

The suggested clearer wording was implemented (lines 274-276).

P8 L233-240: The contributions of the different regions is an important result. To highlight this result even better, I recommend to summarize the contributions of different region in an additional table.

We agree that a table would better summarize the regional contributions, and one has been added (Table 1).

P9 Fig. 3: caption 'Figure 1' -> 'Figure 3'. Please enlarge the height of the panels.

The caption was fixed (lines 310-315), and the figure (Figure 4) reorganized with the panels in a single column to allow for a significant enlargement of all the panels.

P9 L250: --> 'Figure 3 shows the frequency distributions of OTs in 1K intervals for each month related to ...' Yes? Why is the frequency distributions of OTs shown versus the average and maximum potential temperature of OTs and not versus the the potential temperature itself. Please clarify your choice and explain in more detail how the frequency distributions are calculated? It

would be good to show the distance to the tropopause (Fig.3b/d) also in pot. temperature (in K) to make it comparable to Fig. a/c.

OTs detected in this study consist of multiple pixels identified from the Meteosat-8 multispectral imagery. Here, a pixel refers to the minimum spatial unit resulting from the Meteosat-8 horizontal resolution (approximately 4 km). We report OTs as distinct events rather than by pixel count because the temporal resolution (15 minutes) is insufficient to account for the full temporal/spatial evolution of an OT. We characterize each individual OT both by the average potential temperature of its constituent pixels and by the maximum potential temperature reached by its constituent pixels.

As suggested, we recalculated the tropopause relative analysis in terms of potential temperature (Figures 4, 5, 7, and 8; and relevant text).

A discussion of the composition of each OT by individual pixels, and the decision to calculate the distributions by OT rather than by pixel was added to the Data and Methods section (lines 145-160). To the discussion of Figure 4, text identifying the average and maximum potential temperatures as the average of and maximum within the pixels that comprise each individual OT has been added (lines 300-304).

P9 L255/257: '...of the pixels within each OT.' --> please clarify 'pixels'?

See the above response to the referee's comment.

P9 L264: 'As this trend is present in tropopause relative height as well as potential temperature, it is indicative of more vigorous convection rather than simply being a consequence of the seasonal vertical motion of the tropopause.' I would call this not 'trend', better 'shift of the maximum of OTs frequency distributions during summer 2017'. Add the mean pot. temperature of the tropopause for each month in Fig. 3a/c to demonstrate the vertical shift of the tropopause from May until October. What is the variability of the tropopause height during one month? Is the distance to the tropopause calculated for each OT to the tropopause and subsequently the mean distance to the tropopause is calculated or is for each OT the distance to the mean tropopause calculated. Please clarify this point.

The wording has been changed from "trend" to "shift." The sentence (lines 317-323) was also reworded to indicate that the usage of the tropopause relative vertical coordinate shows that the increase in potential temperatures reached by OTs results in further intrusion into the stratosphere, even as the tropopause is higher in potential temperature later in the season.

To illustrate the vertical shift of the tropopause from May through October, supplemental Figure S3 which shows the daily average of the local tropopause potential temperatures associated with each OT has been added. The monthly average tropopause height is lowest in May (370.5 K), rises through August (374.8 K), before declining through October (371.6 K). The standard deviation of tropopause potential temperature across the entire time period is 3.9 K. May has the greatest change in tropopause height within a single month, with a standard deviation of 5.5 K.

The distance to the tropopause is calculated for each individual OT using the local tropopause, and subsequently the mean of these values is calculated. An explanation of this calculation (lines 302-303) has been added to the paragraph that introduces Figure 4.

P10 L269: 'This is visible in the large number of OTs within the highest bins (337 and 572 OTs for average and maximum tropopause height, respectively), which capture all OTs that reach a height above 2.95 km above the tropopause.' What is the maximum height above the tropospause of these extreme convective events? This peak at 3.0 km looks odd. The authors should enlarge the x-axis up to the maximum height above the tropopause (shown in level of pot. temperature).

The distribution of potential temperatures reached by OTs exhibits a long tail, up to one OT with a potential temperature of 493 K (121 K tropopause relative potential temperature). As such, including the full range of the distribution would decrease the visibility of the figure. For this reason, all values outside of the range shown in histogram are included in the uppermost and lowermost bins. Additional explanatory text has been added (lines 326-334), and supplemental Figures S1 and S2 that show the distribution of the long tails not shown explicitly in Figure 4 have been added.

P10 L275: 'seasonal distribution' -> 'intraseasonal distribution' P10 Fig. 4: Similar question as to Fig.3: Why is the frequency distributions of OTs shown versus the average and maximum potential temperature of OT and not versus the the potential temperature itself. Maybe there is a misunderstanding, please clarify. Show in addition the distance to the tropoause in potential temperature coordinates.

See the response to the referee's comment on P9 L250 for an explanation of the frequency distribution analysis. The tropopause relative height coordinate has been updated to potential temperature coordinates for all relevant figures and discussing text.

P11 L96: 'This northward migration is consistent with the expected geographic evolution of the Asian monsoon (Kajikawa et al., 2012; Romatschke et al., 2009)'. The northward movement of the Indian summer monsoon is known for a long time in India. It would be respectful to cite here also some references from Indian colleagues. e.g references in Goswami, B. N.: South Asian monsoon, in: Intraseasonal Variability in the Atmosphere-Ocean Climate System, 2nd edn., chap. 2, edited by: Lau, W. K. M. and Waliser, D. E., Springer-Verlag, Berlin, Heidelberg, 21–72, 2012. I am sure there are more.

To correct this large oversight, we have added the following citations: Abhik et al., 2013; Ganai et al., 2019; Goswami, 2011; and Sikka & Gadgil, 1980 (lines 358-359).

P11 Fig. 5: The individual plots are too small. The numbers on the x-axis are truncated.

Figure 6 (previously Figure 5) has been reorganized with the months as rows rather than columns to allow for a significant enlargement of all the panels and clearer axis labelling.

P11 L311: '... the regions of most frequent cross-tropopause convection are co-located with the most extreme OLR and precipitation values.' For me is the co-location of OTs and OLR not so evident. Please rephrase this sentence.

The sentence has been reworked and expanded upon to add a more nuanced description and interpretation of areas of agreement as well as areas of disagreement (lines 363-371).

P11 L311: 'seasonality' -> 'intraseasonality'

The suggested wording was added (lines 363-365).

P12 L322: 'This is likely due to the lower tropopause in the earlier months' It would be very helpful for the understanding of the presented results and their interpretation to demonstrate the intraseasonal variability of the tropopause height over the Asian monsoon region.

To illustrate the intraseasonal variability of the tropopause from May through October, supplemental Figure S3 which shows the daily average of the local tropopause potential temperatures associated with each OT has been added.

P11 Fig. 5: The individual plots are too small. The titles are truncated. The arrows indicating the horizontal winds in the bottom row are not visible without strong zoom in. Please enlarge all figures.

Figure 6 (previously Figure 5) has been reorganized with the months as rows rather than columns to allow for a significant enlargement of all the panels.

P14 L383: 'seasonal trends' -> 'intraseasonal variability'

The suggested wording was added (lines 451-453).

P15 L387: 'Most of this convection occurs within the Indian subcontinent with North India contributing 29.0% of all OTs, South India contributing 11.7% of all OTs, and the Bay of Bengal contributing 15.2% of all OTs. Together with the Indian Ocean region (19.2%), the most cross-tropopause convection occurs in these regions, and they cumulatively account for 75.1% of all OTs.' That sounds odd and I don't think that the Bay of Bengal is on the Indian subcontinent. --> 'Most of the OTs occurs over South Asia with contributions mainly from North India (29.0%), Indian Ocean region (19.2%), the Bay of Bengal (15.2%) and South India (11.7%) in summary 75.1% of all OTs.' A table listing the contributions of all regions would be very helpful.

The sentence (lines 456-458) has been rephrased as suggested, and a table of regional contributions (Table 1) has been added.

P15 L393: 'maximum height of 387 K corresponding to 1.46 km above the tropopause.' As discussed above I think it is more useful to give the distance to the tropopause in K.

The tropopause relative vertical coordinate has been changed to potential temperature (lines 462-463).

P15 L394: 'seasonal trends' -> 'intraseasonal variability'

The suggested wording was implemented (line 464).

P15 L403: 'In the Arabian Sea, most OTs occur in June, with a "hotspot" corresponding to a single large storm system.' I'm confused in line L374 is written ' is likely sourced from a single

large storm system'. Please clarify. It would strengthen your results if you could demonstrate this is connected (maybe indicating the storm track in your figures) to the storm system or not.

Supplementary Figure S4 and relevant text (lines 440-444) have been added that show the storm track of the large storm system in the first week of June in the Arabian Sea with visible reflectance near 0.6 microns data from the gridded International Satellite Cloud Climatology Project B1 (Knapp et al., 2011).

P15 L403: 'Whether these trends in cross-tropopause convection are recurring features should be explored in future research.' I recommend to remove this sentence. If you draw a connection to future projects, it sounds that your study is incomplete.

The purpose of describing potential future studies is to extend the scientific conversation by indicating new scientific questions that arise from the conclusions of this study that were previously not considered or could not be formulated without the context of this study.

P15 L408 'In contrast to prior work that has emphasized either oceanic (e.g. James et al., 2008) or land-based (e.g. Bergman et al., 2013) convective source regions as dominant, we find that both contribute significant amounts, though with different seasonal distributions.' Please clarify: dominant sources of convection impacting the lower stratosphere / the chemical composition Asian monsoon anticyclone or just the occurrence of convection ?? Further, in the last decades there are several studies analyzing possible convective source regions contributing to the Asian monsoon anticyclone, ATAL or the lower stratosphere over Asia. I remember that both different source regions are identified as well as their intraseasonal variability. The authors should discuss this issue more comprehensively including more references. The authors mention already some references in the introduction.

The sentence has been reworded to clarify that the topic is dominant sources of convection impacting the lower stratosphere. Additional discussion has been added to more comprehensively compare our results to studies which analyzed convective source regions of the Asian monsoon anticyclone. (lines 477-496)

P15 L416-424: I am not sure if this paragraph is an added value for the conclusions, maybe it can be removed or moved to other sections.

A brief description of the limitations of the study can be important context for readers to properly evaluate the implications of the conclusions presented and prevent overstatements.

P15 L425-435: The authors should revise this paragraph and sharpen their main messages.

The concluding paragraph (lines 506-520)has been revised to communicate the main conclusions of the study more clearly.
* * *
minor comments:

P1 L22: 'the contributions across the entire region' -> 'contributions from different regions' ??

Reworded to "contributions across both land-based and oceanic regions." (lines 21-24)

P2 L36: 'are transported' -> 'are transported upwards'

Reworded as suggested. (lines 38-39)

P4 L123: '(see Figure 1c)' -> '(see Figure 1d)'

Corrected.

P5 L139: ITCZ is not introduced

The acronym is defined. (line 183)

P5 L148: 'OLR' -> 'outgoing long-wave radiation' (avoid shortcuts in titles)

Reworded as suggested. (line 194)

P5 L155: 'and time period' -> 'and time period from 1 May to 31 October 2017'

Reworded as suggested. (lines 200-201)

P5 L154: 'We then compare the distribution of cross-tropopause convection with other convective indicators:' -> 'The distribution of cross-tropopause convection is compared with other convective indicators such as ...' (two times in succession 'We..' is used at the beginning of the sentence)

Reworded as suggested. (lines 201-202)

P6 Fig.1: 'the total OTs observed in each region' -> 'the total number of OTs ...'

The caption has been reworded to reflect the separation of the prior Figure 1d into a separate figure. (lines 217-222)

P10 L275: 'Figure 2' -> 'Fig. 2a

Reworded as suggested. (line 335)

**References**

[revised manuscript text omitted]

**Response to Referee 2**

We thank the reviewer for their insightful and thorough criticism. By addressing their comments, we have improved the study as described below. The reviewer's comments are shown in black, and our responses are shown in blue.

Review of C. E. Clapp et al. "Distribution of cross-tropopause convection within the Asian monsoon region from May through October 2017"

The manuscript by Corey Clapp and coauthors provides an analysis of geographic and intraseasonal distribution of cross-tropopause convection in the Asian Summer Monsoon region. The analysis relies on the overshooting tops database spanning a full monsoon season in 2017 and including nearly 41,000 events derived from Meteosat-8 geostationary imagery data. The distribution of overshooting tops is compared with OLR and precipitation data. The study points out significant intraseasonal and spatial variability of tropopause-overshooting convection in the Asian monsoon region.

The study represents a valuable contribution to a better understanding of the source regions and variability of the tropopause-overshooting convection in the Asian monsoon region. With that, the presentation of the results and their current context require careful revision before acceptance to ACP.

General remarks.

• The 2017 Asian monsoon season addressed by the study was the target of StratoClim campaign, which included extensive airborne and balloon-borne measurements across the Southern slopes and the North India regions. There is a large number of studies that followed this experiment, some of which are cited in the paper however this overview is far from being

complete. I suggest that the authors make sure to mention all the relevant papers and carefully discuss the present results in relation with the previous work. More specific remarks on that matter are provided hereinafter.

To put our study more clearly in the context of the StratoClim campaign, a discussion of the results from StratoClim has been added to the introduction (lines 101-114). A discussion of the results of our analysis in relation to the StratoClim studies has been added to the summary and discussion section (lines 493-506).

 • The graphical material is often hard to read, this is particularly the case for the OT maps. I suggest to use a different color map or otherwise make sure that the coastlines are clearly visible in each panel.

The presentation of the figures, including the size and colormap, has been improved to increase visibility. In general, all of the figures were reorganized to allow for significant enlargement.

• I wonder if is the authors could provide an estimate of the total OT area, which would help understanding the magnitude of the impact of tropopause-overshooting convection. Such information could be used to constrain the modeling studies.

In the analysis we chose to report the individual OTs as singular events rather than implement an area/pixel accounting of OTs because the temporal resolution (15 minutes) is insufficient to account for the full temporal/spatial evolution of an OT. With this temporal resolution, it is not possible to verify that the OT area observed at any given time is indicative of the areal coverage of that OT across its entire lifetime relative to the size of other OTs. For this reason, we focus on OTs as representing singular updraft areas that cross the tropopause, regardless of their spatial coverage. Further, at the 15-minute temporal resolution it is highly likely that OTs are missed, given their short time duration. Estimating a full account of the total OT area is a worthwhile endeavor but requires an observational dataset with a time resolution on the order of 30s-1min.

• Given the content of the last section, it should rather be named "summary and discussion"

The suggested section title has been implemented. (line 449)

Specific remarks.

L29 – 37. Here the authors define the study objectives. I would suggest to move it towards the end of the Introduction.

We define the study objectives in the first paragraph as a thesis statement to frame the scientific background of, and motivate research into, convective transport in the Asian monsoon region. We have reworked the final paragraph of the introduction section (lines 115-129), however, to restate these objectives and emphasize the advantages of our analysis more clearly.

L40 – 45. The referencing should be completed with StratoClim studies, such as Brunamonti et al., ACP 2018; Lee et al., 2021; Lamraoui et al., ACPD 2022). In particular, the source regions for the cross-tropopause convection are discussed in detail by Khaykin et al., ACP 2022.

Additional StratoClim studies have been cited with corresponding discussion, and a paragraph discussing the StratoClim in relationship (lines 101-114) to this study has been added to the Introduction section (Brunamonti et al., 2018; Bucci et al., 2020; Johansson et al., 2020; Khaykin et al., 2022; Lee et al., 2019; Lee et al., 2021; Legras & Bucci, 2020; Nützel et al., 2019; Vogel et al., 2019; von Hobe et al., 2021; Yan et al., 2019).

L55 – 57. For the effects of eddy shedding one might refer to Fujiwara et al., ACP 2021. For the transport of Asian pollution towards midlatitudes a pertinent reference would be Khaykin et al., ACP, 2014

The suggested references have been added to the paragraph (Fujiwara et al., 2021; Khaykin et al., 2017). (lines 66-72)

Fig. 1 caption. The description of the panels should be in order

Figure 2 (previously Figure 1) has been significantly reorganized in response to commentary from Referee 1, including ensuring that the panel descriptions are in order.

L.123. Fig 1c -> Fig. 1d ?

The correction has been implemented.

L. 183-185. I am not sure to understand the line of logic here. What is meant by the particular efficiency of TB due to central location?

Here we briefly summarize the findings of Tissier and Legras (2016) and Legras and Bucci (2020). In disagreement with prior studies, they find that the Tibetan Plateau is not a numerically significant contributor to convective transport, but they find that vertical transport over the TP is particularly efficient. Specifically, a high proportion of convection reaching the UT over the Tibetan Plateau is subsequently transported into the LS via slow ascent. In Legras and Bucci (2020), this efficiency is attributed to the fact that parcels entering the AMA over Tibet enter the core region of the AMA, circulate for longer, and therefore experience greater uplift.

In this context, the lack of OT activity over the Tibetan Plateau confirms that convection in this region influences the LS through the slow uplifting process, not through cross-tropopause convection. The agreement with Legras and Bucci (2020) is particularly relevant as their study also covered the 2017 Asian summer monsoon. Additional clarifying text has been added to the relevant paragraph (lines 223-237).

L.191-199. This paragraph is particularly difficult to follow. The term "convective activity" seems to be used for both the OT and OLR features, which renders unclear the discussion on their similarities.

We have reworded the instances of "convective activity" to refer explicitly either to OT or OLR features. (lines 238-247)

L.202 I do now see any significant agreement between OT and precipitation distributions. Overall, I find the discussion that follows largely unclear.

This discussion has been rewritten and expanded (lines 248-259) to discuss specific areas of agreement and disagreement between the OT and precipitation distributions.

Fig. 2. The panels are too small and barely readable. It is nearly impossible to distinguish between the different curves of similar color.

The presentation of the figures, including the size and colormap, has been improved to increase visibility. In general, the figures were reorganized to allow for significant enlargement.

Fig.2 caption. Wrong referencing to Fig. 1

The error in the caption has been corrected. (line 288)

L.220-225 I believe a brief mention on the limited representativeness of OT evolution would be pertinent here.

This limitation, first discussed in the Data and Methods section, has been added. (lines 267-268)

Fig. 3 Where does an isolated feature at highest altitude come from?

The distributions of OT potential temperatures have long tails on the high-value end. In order to capture all OTs without reducing visibility by expanding the x-axis, all OTs that have values above the highest shown value have been placed in the bin with the highest shown value. Additional explanatory text has been added (lines 326-334). Supplemental Figures S1 and S2, which show the distribution of the long tails specifically have also been added.

L. 244. Altitude distribution -> vertical distribution

The correction has been implemented. (line 310)

L.254 redundant with Fig. 3 caption.

The redundant sentence has been removed.

L.396 I do not fully agree with the statement regarding the match between OLR, precipitation and OT, or at least it is not obvious from the figures. Alternatively, if that is indeed the case, does the OT analysis provide an added value for a better characterization of ASM convection

This discussions of comparisons between distributions of cross-tropopause convection and convection more broadly as represented by the OT, OLR, and precipitation distributions (lines 238-259, 358-371, and 380-396) have been rewritten and expanded to discuss specific areas of agreement and disagreement. In these sections, particular attention was given to discussing how differences between these distributions would arise in regions with significant, but strictly tropospheric convection, that did not exhibit OT activity. This sentence has been reworded to reflect that, as a unique subset of extreme convection, the OT distribution had better agreement with the minimum OLR, and maximum precipitation distributions, but not unexpectedly, did not always match these metrics which also included purely tropospheric convection (lines 466-468).

**References**

[revised manuscript text omitted]

---

## Author Response (AR2)

Response to Referee 1

We thank the reviewer for their additional thorough comments. We have addressed the minor corrections as detailed below. The reviewer's comments are shown in black, and our responses are shown in blue.

The manuscript is much improved compared to the previous version and I have only some minor comments, that should be considered before the final publication.

p2, L61: 'subsequent heating results in slow ascent at a rate of approximately 1.0-1.5 K per day within the AMA'

The ascent rate of depends on the used reanalysis. The number of 1.0-1.5 K per day is inferred from ERA-Interim reanalysis. Please check this in literature and make the statement more accurate.

Additional text (lines 61-64) that explains that the slow ascent rate is dependent upon reanalysis derivation with a supporting reference (Tegtmeier et al., 2020), and that the 1.0-1.5 K per day rate refers specifically to a study that utilized ERA-Interim (Legras & Bucci, 2020) has been added.

p3, L70: 'Muller et al., 2016' --> 'Müller et al., 2016'

The error has been corrected (line 73).

p5, L132: 'The study domain from 10°S to 55°N and from 40 to 115°E' -->
'The study domain from 10°S to 55°N and from 40 to 115°E (see Fig. 1)'

The suggested reference to Figure 1 has been included (line 134).

p6 L187: 'seasonal trends' --> 'intraseasonal variability'

The suggested wording was implemented (line 189).

p12 Fig. 3: To avoid any confusion it would be helpful to us in Fig. 3a others colors (e.g black and gray) in contrast to those used in Fig. 3b-d. In general all lines in Fig. 3 should be somewhat thicker.

Figure 3 has been updated with thicker lines in all panels, and the colors of the lines in panel a have been changed to black and gray.

p13 Tab.1: 'percent' --> 'fraction [%]'
'Area normalized percent' --> 'Area normalized [%]'

The headers in Table 1 have been changed as suggested.

p24 L478: 'There has been disagreement in prior work regarding whether oceanic or land-based convective source regions for the Asian monsoon anticyclone are more important.' Please add here to the discussion that differences in used reanalyses contribute to this disagreement.

Additional text noting that differences in reanalyses used across the studies referenced contribute to their disagreement has been added (lines 487-489): "Here, differences between the reanalyses used in modelling studies of Asian monsoon anticyclone convective source regions (e.g. Bergman et al., 2013; and Chen et al., 2012) contribute to the discrepancies amongst their conclusions."

p25 L512: 'Our work also demonstrates that it is critical to consider multi-month time-scales given the large intraseasonal differences in the contributions of each source region. Confining an analysis to a single month would over-emphasize certain source regions while missing others.... '

The intraseasonal variability of different boundary source regions contribution to the Asian monsoon anticyclone is already discussed in Vogel et al. 2015. This reference should be mentioned within the discussion.

Discussion of the results of Vogel et al. (2015) regarding intraseasonal variability of different source regions to the Asian monsoon anticyclone has been added to the conclusion where our results on regional differences are first described (lines 472-474): "These findings are largely in agreement with Vogel et al. (2015), which examined the intraseasonal variability of convective source regions within the Asian monsoon region during 2012 with small differences likely arising from differences in time period considered and in the spatial regions considered."

A reference has also been included in the final paragraph (lines 518-520): "Our work also demonstrates that it is critical to consider multi-month time-scales given the large intraseasonal differences in the contributions of each source region in agreement with Vogel et al. (2015)."

**References**

Legras, B. and Bucci, S.: Confinement of air in the Asian monsoon anticyclone and pathways of convective air to the stratosphere during the summer season, Atmos. Chem. Phys., 20, 11045-11064, 2020.

Tegtmeier, S., Krüger, K., Birner, T., Davis, N. A., Davis, S., Fujiwara, M., Homeyer, C. R., Ivanciu, I., Kim, Y.-H., Legras, B., Manney, G. L., Nishimoto, E., Nützel, M., Kedzierski, R. P., Wang, J. S., Wang, T., and Wright, J. S.: Chapter 8: Tropical Tropopause Layer, SPARC Reanalysis Intercomparison Project (S-RIP) Final Report, No. 10, edited by: Fujiwara, M., Manney, G. L., Gray, L. J., and Wright, J. S., 2020b.

Vogel, B., Günther, G., Müller, R., Grooß, J.-U., and Riese, M., Impact of different Asian source regions on the composition of the Asian monsoon anticyclone and of the extratropical lowermost stratosphere, Atmos. Chem. Phys., 15, 13699–13716, https://doi.org/10.5194/acp-15-13699-2015, 2015.

---

## Author Response (AR3)

**Response to remarks on file validation**

We have updated the color schemes for Figures 2 through 9 to a color scheme that retains information with colorblindness and the corresponding captions. Figure 1 already is interpretable to readers with differences in color perception because of the text labels.